# Constitutive Neurogenesis and Neuronal Plasticity in the Adult Cerebellum and Brainstem of Rainbow Trout, *Oncorhynchus mykiss*

**DOI:** 10.3390/ijms25115595

**Published:** 2024-05-21

**Authors:** Evgeniya Vladislavovna Pushchina, Anatoly Alekseevich Varaksin

**Affiliations:** A.V. Zhirmunsky National Scientific Center of Marine Biology, Far Eastern Branch, Russian Academy of Sciences, 690041 Vladivostok, Russia; anvaraksin@mail.ru

**Keywords:** glutamine synthetase, nestin, vimentin, adult neural stem/progenitor cells, doublecortin, cerebellum, brainstem, *Oncorhynchus mykiss*

## Abstract

The central nervous system of Pacific salmon retains signs of embryonic structure throughout life and a large number of neuroepithelial neural stem cells (NSCs) in the proliferative areas of the brain, in particular. However, the adult nervous system and neurogenesis studies on rainbow trout, *Oncorhynchus mykiss*, are limited. Here, we studied the localization of glutamine synthetase (GS), vimentin (Vim), and nestin (Nes), as well as the neurons formed in the postembryonic period, labeled with doublecortin (DC), under conditions of homeostatic growth in adult cerebellum and brainstem of *Oncorhynchus mykiss* using immunohistochemical methods and Western Immunoblotting. We observed that the distribution of vimentin (Vim), nestin (Nes), and glutamine synthetase (GS), which are found in the aNSPCs of both embryonic types (neuroepithelial cells) and in the adult type (radial glia) in the cerebellum and the brainstem of trout, has certain features. Populations of the adult neural stem/progenitor cells (aNSPCs) expressing GS, Vim, and Nes have different morphologies, localizations, and patterns of cluster formation in the trout cerebellum and brainstem, which indicates the morphological and, obviously, functional heterogeneity of these cells. Immunolabeling of PCNA revealed areas in the cerebellum and brainstem of rainbow trout containing proliferating cells which coincide with areas expressing Vim, Nes, and GS. Double immunolabeling revealed the PCNA/GS PCNA/Vim coexpression patterns in the neuroepithelial-type cells in the PVZ of the brainstem. PCNA/GS coexpression in the RG was detected in the submarginal zone of the brainstem. The results of immunohistochemical study of the DC distribution in the cerebellum and brainstem of trout have showed a high level of expression of this marker in various cell populations. This may indicate: (i) high production of the adult-born neurons in the cerebellum and brainstem of adult trout, (ii) high plasticity of neurons in the cerebellum and brainstem of trout. We assume that the source of new cells in the trout brain, along with PVZ and SMZ, containing proliferating cells, may be local neurogenic niches containing the PCNA-positive and silent (PCNA-negative), but expressing NSC markers, cells. The identification of cells expressing DC, Vim, and Nes in the IX-X cranial nerve nuclei of trout was carried out.

## 1. Introduction

The study of the neuronal development stages in different parts of the fish brain not only elucidates the basic mechanisms of adult neurogenesis but also contributes to understanding the formation patterns in certain brain regions and the involvement of adult neural stem/progenitor cells (aNSPCs) in the construction of the definitive brain structure [1,2]. Previously, the distribution of aNSPCs, formed during the homeostatic neurogenesis in the telencephalon and tectum of the brain, was studied in 3-year-old rainbow trout, *Oncorhynchus mykiss*, in order to address the question concerning the mode of adaptation of the radial glia (RG) structures and the population of neuroepithelial (NE) cells to the continuous brain growth characteristic of salmon [3].

Previous reports have shown that the brain of teleosts continues to grow throughout their life [4]. The brain growth patterns have been described in four European cyprinid fish species (Cyprinidae, Teleostei): roach (*Rutilus rutilus*), bream (*Abramis brama*), common carp (*Cyprinus carpio*), and saber carp (*Pelecus cultratus*) [4]. The neurogenesis of the brainstem of brown trout *Salmo trutta fario* and medaka *Oryzias latipes* [5], adult stem cells in the cerebellum of kinefefish *Apteronotus leptorhynchus* [6], and adult neurogenesis in zebrafish *Danio rerio* including the rhombencephalic region [7] have also been described.

The study of the properties of aNSPCs using a salmon model provides new data on the organization of neurogenic zones in various parts of the brain containing the adult-type stem cells, with focus on their development, origin, cell lines, and proliferative dynamics [8,9]. Currently, the molecular signatures of these populations during homeostasis and repair in the vertebrate forebrain are only beginning to be studied [2,10,11,12]. Outside the telencephalon, the regenerative plasticity of adult stem/progenitor cells and their biological significance remain poorly understood. This is especially true for the caudal parts of the brain and in the brainstem and cerebellum, in particular, where the neural stem cells (NSCs) were found in zebrafish [2,6], as well as in juvenile chum salmon [8] and masu salmon [9]. 

The presence of NSCs in the cerebellum and medulla oblongata of fish differs significantly from adult mammals, in which such foci of homeostatic neurogenesis are not preserved. Previous studies on juvenile chum salmon and masu salmon have shown that the cerebellum is dominated by neuroepithelial cells (NECs) located in various parts of the cerebellar body, dorsal matrix zone (DMZ), and granular eminence [13,14].

RG cells play a central role in the origin and placement of neurons [15] because their processes form structural scaffolds that guide and facilitate neuronal migration. In addition, glutamatergic signaling in the RG of the adult cerebellum (Bergmann glia) of zebrafish is critical for precise motor coordination [16]. Studies in zebrafish have shown that RG cells exhibit spontaneous calcium activity, and functional interaction propagates calcium waves. However, the origin of calcium activity in relation to the ontogeny of cerebellar radial glia is just beginning to be explored, and results in zebrafish have elucidated important aspects of RG organization, morphology, and calcium signaling, emphasizing its role in complex behavioral paradigms [16].

Other studies also indicate that the behavioral activity of zebrafish can be regulated not only by neurons of the medulla oblongata but also by RG cells [17]. In optogenetic studies, it was found that with the transgenic introduction of neuron- and glio-specific vectors into the medulla oblongata of zebrafish and subsequent behavioral experiments with virtual reality simulation and swimming, after unsuccessful swimming attempts, calcium activity in the noradrenergic neurons of the medulla oblongata ceases and corresponding calcium activity appears in RG cells. This work established that the involvement of the RG is not limited to structural and neurogenic properties in adult fish but apparently also extends to the modulation of behavioral activity, in particular swimming [17]. Since these observations are associated with the functional activity of the caudal part of the brain, medulla oblongata, it is very interesting to study the distribution of aNSPCs and in particular RG, corresponding to the adult-type NSCs in this part of the brain in adult trout. Thus, the questions of whether neuronal precursors are preserved in the cerebellum and medulla oblongata of adult trout and which cellular phenotypes these forms are represented by determine significant interest for studying the neuronal plasticity features in the caudal regions of the brain of adult trout. 

The high regenerative and proliferative capacity of the juvenile salmon brain suggests that most aNSPCs are probably multipotent, as they are capable to replace almost all cell lines lost through damage, including NE cells, radial glia, oligodendrocytes, and neurons. To date, this hypothesis has been supported to a large extent by studies on quiescent Müller retinal glia in adult humans [9] and *Danio rerio* [11,18,19]. However, the unique regenerative profile of certain cell phenotypes in heterogeneous salmon brainstem cell niches is still unclear.

Different types of neuronal progenitors, as was found earlier, are present in sufficient numbers in different parts of the juvenile Pacific salmon brain [8,20]. Study of aNSPCs’ properties in dynamics, in adult animals, and through various post-injury periods, when the initial potential of aNSPCs and the probability of their involvement in the homeostatic or reparative processes can be determined, is of particular relevance [12,20,21]. In the primary periventricular areas of the tectum, tegmentum, telencephalon, and cerebellum of juvenile chum salmon, *Oncorhynchus keta*, a high proliferative activity of cells was observed on days 3–7 post-injury [8,9,22]. To date, the biological mechanisms associated with the high neuroreparative potential of juvenile Pacific salmon remain unknown. However, the significant body size that Pacific salmon (chum, sockeye, and coho salmon) reach during feeding in the ocean [23] suggests that the increase in the size of muscle tissue is controlled with the involvement of many neurons formed throughout the life cycle [18,24]. 

Pacific salmon belong to a group of ray-finned fishes, a branch of which emerged on the evolutionary tree rather recently, approximately 400 million years ago. One of the most intriguing features of the salmon family is the full-genome duplication event, designated as SsR4 or WGD4, which occurred approximately 70 to 80 million years ago, according to [25]. This event has occurred at various times, ranging from 96 ± 5.5 million years ago to the present day [26]. Genome-wide duplication plays a significant role in the evolution of vertebrates, particularly in the fish species. It contributes to the development of new functional capabilities by copying not only structures but regulatory regions of genes as well. At the same time, half of the resulting tetraploid genome continues to control previous functions, which support current adaptations, while the other part is freed from those functions and can rebuild quickly. This process is not limited by stabilizing selection, which determines the gene complex differentiation and evolutionary adaptation to the environment, and directly affects speciation processes [27]. 

Currently, most studies associated with Pacific salmon research address various ecological and ichthyological problems [23] and, to a much lesser extent, the issues of physiological regulation, mediator-modulatory, and reparative properties of the CNS. The results of studies on the regenerative potential of aNSPC populations in neurogenic zones outside the forebrain in adult zebrafish [28,29,30] have shown that certain subtypes of aNSPCs are characterized by different regenerative abilities and different molecular control mechanisms. 

It is still unclear whether these properties are the result of cell-specific repair programs or any effect of the microenvironment surrounding stem cells in case of damage. It was previously shown that inactive astrocytes in physiological conditions can be converted into neurons in vitro and in vivo by adjusting the level of Notch signaling or forced expression of proneural factors such as Neurod1 and, thus, can serve as a source of neuronal recovery after acute traumatic brain injury [31,32]. In addition, populations of ependymal cells in the forebrain and spinal cord of adult mammals also have a regenerative potential for neuron formation [32,33,34,35,36,37,38,39,40,41,42,43]. In contrast to mammalian models, we consider the rainbow trout brain as a preferred vertebrate model to extend our understanding of the cellular and molecular programs required for successful CNS growth and regeneration. A distinctive feature of the salmon brain is fetalization, or the presence of signs of the embryonic brain structure in juvenile *Oncorhynchus keta* [8], *Oncorhynchus masou* [9], and adult *Oncorhynchus mykiss* individuals [3]. In this regard, the study of the expression of glutamine synthetase (GS), vimentin (Vim), and nestin (Nes), which are molecular markers of aNSPCs in zebrafish [11,12,24,28] and *Apteronotus leptorhynchus* [35], contributes to extending our knowledge of the nature of aNSPCs in less studied fish models and, therefore, is important for the comparative study of neurogenesis in general. 

The patterns of the IHC activity of molecular markers of neurogenesis of Pacific salmon juveniles are quite complex, and for the verification of cellular localization, given the complexity of the organization of the caudal parts of the brain and weak anatomical compliance with the results of immunolabeling on other model objects, in particular *Danio rerio* [11,12,15,18,19], immunolabeling is a necessary stage of the study of the aNSPCs’ morphology.

It should also be taken into account that in the brains of juveniles and more so in adult individuals of Pacific salmon there is a huge number of NE precursors associated with the extensive creation of new neurons in the growing brain [8]. The proportion of such cells is significantly higher than in other fish species, in particular, *Danio rerio*, which determines large differences in the patterns of localization and the presence of constitutive neurogenic zones containing NE precursors. The fetalization of the brain—the preservation of signs of embryonic structure in salmon fish at more adult stages of development and in adult animals in general—puts salmon in a special position and determines the presence of special signs in their brain that differ from other fish species. Our study objective was to assess the distribution of proliferative activity and GS, Vim, and Nes, as well as the DC immunolabeling of neurons formed in the postembryonic period, in the cerebellum and brainstem of adult rainbow trout, *Oncorhynchus mykiss*, under conditions of homeostatic growth.

## 2. Results

### 2.1. Proliferating Cell Nuclear Antigen (PCNA) in Trout Cerebellum

As a result of labeling the proliferating cell nuclear antigen in the molecular layer of the corpus cerebellum (CC), single PCNA+ cells were detected both in the surface layers and in the parenchyma (Figure 1A). Deep in the ML, there were PCNA+ cells forming small clusters, single intensely labeled PCNA small cells of a rounded morphology (Figure 1B), and single PCNA+ fibers of a radial glia (Figure 1B). Single PCNA+ fibers of RG are not routine observations with PCNA immunohistochemistry. PCNA is indeed a nuclear antigen, and therefore, PCNA specific immunoreactivity usually consists of labeled nuclei. The RG in the vertebrate brain is an exception to this rule, as its radial fibers express PCNA immunoreactivity. Previously, data on the presence of PCNA+ RG were shown in studies on the post-traumatic reaction in the brain of adult trout [36].

Table 1 shows the morphometric and densitometric parameters of PCNA+ cells, as well as topographic areas of cell localization in the cerebellum and brainstem (Table 1). In the surface layers of the ML, single clusters of PCNA+ cells with rounded and elongated morphology were detected (Figure 1C). Clusters of PCNA+ cells were localized in the subependymal zone (Figure 1C). Rounded and elongated PCNA+ cells were detected in the DMZ of the trout cerebellum (Figure 1D, Table 1). The basal part of the DMZ was dominated by PCNA+ cells (Figure 1D). An extensive population of PCNA+ cells was identified in granular eminences (GrEms) (Figure 1E, Table 1). Immunopositive cells formed a multilayered structure in the surface part of the GrEm (Figure 1E,F). Clusters of PCNA+ cells were found in the submarginal zone (Figure 1E, Table 1); in the deep layers, the distribution density of PCNA+ cells was lower (Figure 1F).

A comparative analysis of the distribution of PCNA+ cells in various areas of the trout cerebellum showed heterogeneity in the distribution of proliferating cells in various areas of the molecular layer, as well as PCNA+ clusters and parenchyma of the molecular layer and granular eminences (Figure 1G).

### 2.2. PCNA in the Trout Brainstem

In the brainstem at the level of the isthmus nucleus (NIs), PCNA immunolocalization was detected in the periventricular zone (PVZ, Figure 2A, Table 1). Immunopositive cells were present as undifferentiated intensely labeled rounded cells (Figure 2A, Table 1) and elongated, intensively or moderately labeled cells (Figure 2B, Table 1), as well as two types of RG (data not shown in Figure 2, Table 1). In the region of the dorsal reticular formation (DRF, Figure 2C), the PVZ was dominated by elongated cells (Figure 2C) whose distribution density was lower than in the tegmental zone. In the SMZ, some PCNA+ cells had radial outgrowths (Figure 2C,D, Table 1) and apparently represented a population of slowly proliferating RG cells. The intensity of PCNA+ RG immunolabeling varied from intense to moderate (Figure 2C,D, Table 1). In the deep parenchymal layers of the brainstem at the level of the DRF, single intensely labeled PCNA cells were detected (Figure 2C, Table 1). In some subventricular regions, at the level of the nucleus of the VII pair of cranial nerves, aggregations of PCNA-negative cells in the PVZ, forming constitutive local clusters, were detected (Figure 2E). In the PVZ at the level of such clusters, PCNA-immunopositive cells were not usually detected (Figure 2F). However, PCNA+ fibers of RG were observed (Figure 2F). Thus, PCNA+ cells with neuroepithelial morphology corresponding to embryonic-type NSCs, as well as slowly proliferating PCNA+ radial glia corresponding to aNSPCs, were identified in the adult trout brainstem.

A comparative analysis of the distribution of PCNA+ cells in various areas of the trout brainstem showed heterogeneity in the distribution of proliferating cells in various areas of the PVZ and superficial layer of the brainstem, as well as PCNA+ clusters and parenchyma of the brainstem (Figure 2G).

### 2.3. Glutamine Synthetase in the Trout Cerebellum

Glutamine synthetase (GS) is a specific glial protein that converts toxic extracellular L-glutamate to glutamine, a neutral amino acid [37]. Under normal conditions, this mechanism prevents the neurotoxic accumulation of glutamate in neural tissue, protecting neurons from cell death. In fish, GS often labels radial glial cells [3] but can also be detected in glia-like cells [9]. In the *A. leptorhynchus* cerebellum, the amount of GS increases after injury [38], while the synthesis of this enzyme in the mammalian CNS decreases after injury [37]. The presence of GS-positive (GS+) fibers of RG in the norm indicates persistent, postembryonic neurogenesis and suggests GS to be one of the morphogenetic markers of RG [39].

In the cerebellum of rainbow trout, *O. mykiss*, glutamine synthetase (GS) was detected in various types of cells and fibers in the dorsal, lateral, and basal zones of the corpus cerebellum (CC) (Table 2). 

In the dorso-lateral region (Dl) of the CC (pictogram in Figure 3A), several types of GS+ cells were identified in the periganglionic part of the molecular layer (Figure 3A, red arrows). A high intensity of GS labeling was detected in small undifferentiated cells (Figure 3A, black inset) and also in 2–3 elongated cell types (Figure 3A, red inset). In the ganglion layer (Gl), however, no immunopositive cells were found (Figure 3A, white arrows). Larger GS+ cells in the Dl located in the basal part of the molecular layer (ML) had high or moderate immunolabeling intensity (Figure 3B, Table 2). 

The dorsal matrix zone (DMZ) showed high GS activity (Figure 3C, pictogram). In the DMZ, two types of small and one type of elongated intensely labeled GS+ astrocyte-like cells were identified (Figure 3C, red arrows, Table 2). In the basal part of the DMZ GS+, type 1 cells were grouped into small clusters (Table 2, Figure 3D, outlined by red dotted lines). In the lateral part of the DMZ GS+, elongated type 2 cells were located diffusely (Figure 3D, inset), or single intensely labeled cells of the neuroepithelial (NE) phenotype were encountered (Figure 3D, black arrow). In the basal part of the CC GS+, the cellular composition was generally similar to that of the Dl (Figure 3E) with, however, larger astrocyte-like cells dominating (Table 2). At the base of the basal zone (BZ), elongated, GS+, type 1 and 2 astrocyte-like cells formed dense extensive groups (Figure 3E, black inset), while in the middle part of the BZ, the number of immunopositive type 2 and 3 astrocyte-like cells and the intensity of GS+ immunolabeling decreased markedly (Table 2, Figure 3E, red inset). In the baso-lateral zone (Bl), the GS distribution and cell composition (Table 2, Figure 3F) generally matched those of the Dl and BZ (Figure 3F, inset). 

The number of GS+ astrocyte-like cells in the Bl significantly exceeded that in other zones of the CC (Figure 3G, black box). Unbranched GS+ fibers were observed in the ML (Figure 3G, blue inset) with thickening along the fibers (Figure 3G, white arrowheads). In some cases, small clusters of GS+ cells of the NE phenotype were detected in the Bl (Figure 3G, red inset), located near clusters with undifferentiated GS+ cells in the ML (Figure 3G, outlined by white and blue dotted lines). A high-density distribution of GS+ fiber endings was found in the apical part of the ML Dl of the CC (Figure 3G, brown rectangle). In the Dl, similar patterns of distribution of undifferentiated GS+ cells (white arrowheads) along RG fibers were also observed deep in the ML (Figure 3H, red inset). GS+ cells of types 1, 2, and 3 in the Dl were usually located in the basal part of the ML (Figure 3I, black inset) or, in some cases, embedded in the granular layer (GL) (Figure 3I, red arrow). 

The intensity of immunolabeling of elongated GS+ astrocyte-like cells in the Gl area varied from high to moderate (Figure 3I, red arrows). The patterns of distribution of extensive aggregations of undifferentiated GS+ cells in the ML (Figure 3J, red inset) and fibers (Figure 3J, red dotted box) were most typical in the Dl, while the bodies of elongated GS+ astrocyte-like cells of types 1, 2, and 3 were located at the base of the ML (Figure 3J, red arrows). In the baso-lateral periventricular part of the CC (Figure 3K, pictogram), a heterogeneous composition of GS+ cells was observed (Figure 3K, black inset, Table 2). In the medial periventricular zone (Figure 3L, pictogram), such cells often formed local clusters (outlined by red dotted line) of intensely labeled undifferentiated GS+ cells (Figure 3L, black inset).

A quantitative analysis of the total distribution of GS+ cells in different anatomical regions of the cerebellum showed a maximum number of astrocyte-like cells in the Bl and a minimum in the lateral zone (Figure 3M). Significant intergroup differences (*p* < 0.05) in the distribution of GS+ cells between different zones of the CC were revealed (Figure 3M). The comparative distribution of GS+ cells and RG in the same areas of the CC and the data of the intergroup analysis of variance (ANOVA) are shown in Figure 3N.

### 2.4. Glutamine Synthetase in the Trout Brainstem

Throughout the brainstem, the caudal to dorsal tegmental nuclei, and the oculomotor nuclear complex, GS immunolocalization was detected in the periventricular zone (PVZ), interfascicular zone (IFZ), submarginal zone (SMZ), and also around the largest tracts of the brainstem (Figure 4A, Table 2). 

The highest intensity of GS immunolabeling was found in PVZ cells that formed a dense periventricular layer of RG cell bodies whose processes extended in the radial direction and penetrated the brainstem in the form of a dense radially oriented system of fibers, enveloping the descending tract of trigeminal nerve (DTN), lateral longitudinal fascicle (fll), and some other tracts of the brainstem (Figure 4A). Immunonegative vessels were encountered in the PVZ (Figure 4A, blue arrow). 

The largest population of brainstem GS+ astrocyte-like cells was located in the IFZ (Figure 4B, blue square). The IFZ is an intermediate zone of the reticular formation located between the medial longitudinal fascicle (flm) and the fll. The IFZ contained both GS+ astrocyte-like cells and RG fibers (Figure 4B, blue square). Two types of GS+ astrocyte-like cells were identified: moderately or intensely labeled (Table 2). Undifferentiated GS+ cells in the IFZ, as a rule, were located along the GS+ fibers of the RG, and the cell bodies of the RG cells were located in the PVZ (Figure 4C). Undifferentiated GS+ PVZ cells (Figure 4B, red arrowheads) most frequently occupied a periventricular position bordering the IV ventricle. Large RG fibers (Figure 4B, yellow arrows) were identified in the dorso-medial part of the brainstem. In the PVZ, two types of GS+ cells and two types of RG cells (larger, moderately or intensely GS-immunolabeled) were detected (Table 2). All cell types in the PVZ were layered in the periventricular region (Figure 4C, red inset); undifferentiated and elongated type 1 astrocyte-like cells occupied the border region (Figure 4C, red arrowheads), and RG cell bodies of types 1 and 2 were located in deeper layers (Figure 4C, yellow arrows). 

The intertrigeminal zone (ITZ) bundle, which also contained GS+ RG fibers (Figure 4D, blue inset), extended in the dorso-lateral direction from the DTN. Along with intensely immunolabeled fibers (Figure 4E, yellow arrows), undifferentiated, intensely labeled GS+ cells were located in the ITZ (Figure 4E, white arrows) along the RG fibers. Single fibers with varicose thickenings extended dorsally to the ITZ (Figure 4E, black inset). An extensive population of undifferentiated and elongated GS+ cells and radial glial cells was found near the outer surface of the brainstem (Figure 4F, yellow arrows). 

Clusters of single elongated GS+ astrocyte-like cells (Figure 4G, black inset, red arrows) and similar GS+ cells forming small clusters were found in the paramedian region of the brainstem, near the flm (Figure 4G, black inset, red dotted oval). The main motor tracts of the brainstem, including the flm (Figure 4G, yellow arrows) and fll (Figure 4H, yellow arrows), were surrounded by GS+ RG fibers reaching considerable length and thickness (Figure 4H, black arrowheads). Numerous GS+ RG fibers were also identified in the brainstem SMZ (Figure 4I, yellow arrows). In the ventro-lateral region of the brainstem SMZ, GS+ cells (Figure 4J, red arrowheads) and thicker RG fibers (Figure 4J, yellow arrows) were located sparsely, while in the lateral region (Figure 4K, blue inset), on the contrary, they were denser. Diffuse patterns of distribution of similar GS+ cells and fibers were revealed in the medial and paramedian regions of the brainstem SMZ (Figure 4L). 

The results of a one-way ANOVA of the distribution of GS+ cells in different topographic regions of the brainstem are shown in Figure 4M. Significant intergroup differences (*p* < 0.05) were found between the PVZ, IFZ, ventro-lateral brainstem, and ITZ. The ratios of GS+ cells and RG in different areas of the brainstem (M ± SD) are shown in Figure 4N. The data of a quantitative analysis indicate the dominance of GS+ cells compared to RG in all the topographic areas of the brainstem.

### 2.5. Doublecortin in the Trout Cerebellum

Doublecortin (DC) is a microtubule-associated protein expressed by newly generated and migrating neurons; it is usually involved in the neurogenesis processes. Its expression may also be important for the axonal growth or synaptogenesis, a phenomenon that continues in adult neurons. DC is localized both in the cytoplasm and in the nucleus. Its presence is characteristic of most intracellular processes [40], and it is also found in the dendritic growth cone. Expressed during neurogenesis in adult animals, DC is detected during the migration of some cell types [41]. The DC expression persists in postmitotic neurons also and coincides in time with the expression of calretinin [42]. It is hypothesized that in mature neurons DC expression may be re-induced and is required for neuronal plasticity.

In the trout cerebellum, doublecortin immunolocalization was identified in heterogeneous cell populations of different anatomical regions of the corpus cerebellum (CC) located in the Ml, Gl, and granular layer (Grl) (Figure 5A–P, Table 3). In the baso-lateral region of the CC, elongated and oval intensely labeled DC+ cells (Figure 5A, red arrows) were identified in the Gl (Figure 5A, inset, Table 3). The results of the negative control are provided in Appendix A. No structures were found in the corpus cerebellum in the negative control of DC+ (Appendix A). Larger fusiform eurydendroid cells (EDCs) were located singly and, unlike oval and elongated neurons, did not form clusters (Figure 5A, red arrow). In the baso-medial region of the CC, the distribution patterns of DC+ neurons in the Gl included both type 1 and type 2 cells with intensely labeled cytoplasm, as well as larger moderately labeled EDCs and pear-shaped cells (PCs), with a centrally located immunonegative nucleus (Figure 5B, inset, Table 3).

Most DC+ neurons in the basal region were located in the Gl with, however, single ectopic EDCs found located in the Grl (Figure 5C, inset). In the medio-basal part of the CC, the ML formed an invagination in the Grl (Figure 5D), with DC+ neurons of types 1 and 2 detected at the border (Figure 5D, black inset), which were also characteristic of other areas of the basal part of the cerebellum (Figure 5D, blue inset, Table 3). In the ML of this and other regions of the basal part of the CC, numerous small DC+ neurons were found (Figure 5D). In the central region of the basal part (BC) of the cerebellum, a high distribution density of oval-shaped DC+ PCs with intense or moderate labeling (Figure 5E, black inset) was detected along with diffusely located moderately labeled DC in the Grl (Figure 5E, Table 3). A DC+ PC population with heterogeneous immunolabeling intensity (from moderate to high) was identified in the paramedian region of the basal part of the CC (Figure 5F, black inset). Thus, in the basal part of the CC, numerous adult DC+ neurons in the Gl, Grl, and Ml, including neurons that form extracerebellar projections (EDCs and PCs), were identified.

In the dorsal part of the trout cerebellum, several types of DC+ neurons were identified (Figure 5G, red arrows, Table 3). Small, intensely labeled DC+ cells of two types were identified in the dorsal matrix zone (DMZ) of the cerebellum (Figure 5G, inset, Table 3). Undifferentiated small type 1 cells formed dense clusters in the DMZ, while larger elongated type 2 cells were localized in pairs in the ventral part of the DMZ in Grl or formed chains of several cells (Figure 5G, inset). 

Dense aggregations of DC+ undifferentiated type 1 small cells were found in the apical zone of the Ml (Figure 5G, inset, white dotted oval), and some DC+ cells of a similar morphology were diffusely distributed in the deeper layers of the Ml (Figure 5G, inset, yellow arrowheads). In the dorsal part of the CC, outside the DMZ, several types of DC+ cells were identified in the Gl with different immunolabeling intensities (Figure 5H, Table 3). In the Ml of the dorsal part of the CC, small moderately DC+ cells were identified (Figure 5H, orange arrowheads) located along the weakly labeled DC+ radial fibers (Figure 5H, blue arrowheads). 

In the lateral zone of the CC, a heterogeneous population of DC+ cells was identified in the Ml, Gl, and Grl (Figure 5I, Table 3). Moderately labeled RG fibers (Figure 5I, blue arrows) and elongated moderately/intensely labeled DC+ type 2 cells (Figure 5I, orange arrows) were found deep in the Ml. Small undifferentiated DC+ cells (Figure 5I, pink arrows) formed diffuse and denser aggregations in the basal part of the Ml (Figure 5I, black inset, Table 3).

Groups of oval and pear-shaped Purkinje cells (PCs) with moderate DC labeling were identified in the Gl (Figure 5I, red dotted oval, Table 3); such clusters sometimes contained intensely labeled cells (Figure 5I, pink dotted oval). In the Grl of the lateral region, there were single large polygonal DC+ cells with an intense/moderate level of immunolabeling and an immunonegative nucleus (Figure 5I, pink arrows). In some cases, large bipolar, ectopic, moderately labeled EDCs were identified in the Grl (Figure 5J, red arrow), and intensely/moderately labeled type 1 PCs were identified in the Gl of this region (Figure 5J, black inset, Table 3). Large, intensely/moderately DC-labeled, type 2 PCs were detected in the Gl and Grl of the lateral region (Figure 5K, red arrows, Table 3). In the Grl of the dorso-lateral part of the CC, numerous oval and polygonal DC+ cells of types 1 and 2 with moderate or intense labeling were detected (Figure 5L,M, pink arrows, Table 3). In the region of the granular eminence (GrEm), a heterogeneous population of DC+ cells (Figure 5N, red and pink arrows) and fibers was identified (Figure 5N, blue and yellow arrows, Table 3). 

In the basal part of the GrEm, large polygonal DC+ neurons (Figure 5N, black inset, Table 3) and oval moderately labeled DC+ cells of types 1 and 2 (Figure 5N, pink arrows, Table 3) were observed. In the lateral zone of the GrEm, type 1 oval cells were moderately labeled (Figure 5O, pink arrows, Table 3), while type 2 oval cells tended to be intensely/moderately labeled (Figure 5O, black inset, Table 3). In the negative control, no DC+ cells were detected in the GrEm (Appendix A). Single weakly labeled DC+ fibers were also found here (Figure 5O, blue arrows). In the superficial (Figure 5P, red inset) and subsurface (Figure 5P, white dotted inset) layers of the dorso-lateral zones of the GrEm, dense aggregations of intensely labeled elongated cells were found (Figure 5P, black arrows).

The results of a one-way ANOVA showed a comparative distribution of DC+ cells in different topographic areas of the CC (Figure 5Q). Significant intergroup differences (*p* < 0.01) were found between the DMZ and the Grl and also (*p* < 0.05) between the dorsal, lateral, baso-medial (BM), baso-central (BC), and baso-lateral (BL) regions, GrEm, and Grl (Figure 5Q). An estimation of the ratio of the number of DC+ cells to GS+ RG in the basal, lateral, and dorsal regions and the DMZ showed significant differences (*p* < 0.01) in the dorsal region and also (*p* < 0.05) in the lateral region (Figure 5R). A comparative analysis of the distribution of DC+ and GS+ cells did not reveal significant differences in the distribution of immunopositive cells (Figure 5S). However, the estimation of the ratio of DC+ cells and the total number of GS+ cells and RG showed significant differences (*p* < 0.01) in the lateral and dorsal regions and also (*p* < 0.05) in the basal region (Figure 5T).

### 2.6. Doublecortin in the Trout Brainstem

The DC immunolabeling in the trout brainstem revealed various populations of cells formed in the postembryonic period, which included cells belonging to the dorsal somatosensory column: nuclei VII, IX, and X of cranial nerve pairs (Figure 6A, red box), as well as dorsal, interfascicular (Figure 6A, blue square), ventro-lateral (Figure 6A, white dotted oval), and reticulospinal parts of the reticular formation (Figure 6A, Table 3). The results of the negative control of DC immunolabeling in the trout brainstem are provided in Appendix A. In the lateral (Appendix A) and medial parts of the brainstem, no DC+ structures were detected around the flm in the negative control (Appendix A). 

The nucleus of the vagus nerve (NX) (Figure 6B, black box) included four types of moderately/intensely labeled DC+ cells and was located in the dorsal somatosensory column. The nucleus of the X nerve covers the motor and sensory parts. The motor part, containing the parasympathetic column, contains larger perikarya (Figure 6B, black square) than the sensory part, i.e., the nucleus tractus solitaris (NTS) consisting of relay neurons and interneurons (Figure 6F). In this case, the sensory NTS is located more dorsally and closer to the IV ventricle (Figure 6F) than the motor part of the nucleus of the X nerve. The nucleus of the glossopharyngeal IX nerve (Figure 6C, black rectangle) contains three types of moderately labeled DC+ cells. DC+ interneurons (Figure 6D, red arrows) of the IFZ containing moderately and intensely labeled cells were found at the border between the flm and fll (Figure 6D, black inset, Table 3). The NX was located dorso-laterally of the fll (Figure 6D, red box). DC+ neurons of the IFZ were represented by two types of moderately labeled small oval cells and three types of moderately/intensely labeled polygonal cells (Figure 6F, Table 3). Among the DC+ somatosensory neurons of nuclei IX-X of cranial nerve pairs (Figure 6F, blue arrows), moderately labeled RG fibers were detected (Figure 6F). 

The nucleus of the facial VII nerve contained seven types of moderately/intensely labeled polygonal DC+ neurons (Figure 6G, blue arrows) in the somatosensory cell column (Figure 6G, Table 3). In the dorsal reticular formation (DRF), DC+ neurons were more homogeneous (Figure 6H, blue arrows) and were represented by four cell types (Table 3). The smallest undifferentiated cells were intensely DC-labeled; the larger elongated and oval type 1, 2, and 3 cells were intensely or moderately/intensely labeled (Figure 6H, black inset, Table 3). 

The periventricular zone (PVZ) contained undifferentiated and elongated intensely DC-labeled cells (Figure 6I, Table 3), and the subventricular zone (SVZ) contained weakly labeled cells (Figure 6I, red inset). In the PVZ, most of the small undifferentiated DC+ cells were located at the border with the IV ventricle (Figure 6J, red arrows), and moderately labeled RG fibers (Figure 6J, black arrows) spread into the deeper subventricular layers, among which a few intensely labeled cells were found (Figure 6K, black inset). 

A large population of heterogeneous DC+ cells was represented by reticulospinal cells (RSCs) (Figure 6L, black and red insets). RSCs included two types of intensely labeled elongated and oval cells and seven types of polygonal moderately/intensely labeled interneurons (Figure 6L,M, black insets, Table 3). Heterogeneous ventro-lateral cells (VLCs) of reticular formation were localized in the ventro-lateral part of the brainstem (Figure 6M,N, red arrows). The VLC population was represented by undifferentiated intensely labeled cells, two elongated intensely/moderately labeled cell types, intensely labeled oval cells, and two intensely/moderately labeled polygonal cell types (Figure 6M,N, Table 3). Populations of DC+ reticulospinal interneurons (Figure 6O, red arrows) formed discrete functional topographic modules (Figure 6O, red and black insets) within the reticular formation. The complex structure of primary and secondary DC+ interneurons within the area of RSCs (Figure 6P, red arrows) suggests a multilevel hierarchy system associated with the control of motor activity and innervation of the axial muscles. 

The study of the proportion of the number of DC+ cells in various structures of the brainstem by one-way ANOVA revealed significant intergroup differences (*p* < 0.05) between the PVZ and the VLC area and also between the DRF and the RSC area (Figure 6Q). The comparative analysis of the ratio of DC+ cells and GS+ RG showed no significant intergroup differences (Figure 6R). The study of the comparative distribution of DC+ cells and GS+ NE cells revealed significant intergroup differences (*p* < 0.05) between the IFZ and the VLC populations (Figure 6S). A comparative analysis of the distribution of the total number of DC+ and GS+ cells revealed significant intergroup differences (*p* < 0.01) between the IFZ and the VLC populations (Figure 6T).

### 2.7. Vimentin in the Trout Cerebellum

Vimentin (Vim) is a type III intermediate filament protein expressed by astroglial cells [43]. It is commonly regarded and used as a marker of astrocytic glia in the vertebrate brain [44]. Along with nestin, it constitutes a part of the aNSPC cytoskeleton. Immunocytochemical studies of ependymal cells and radial glia, which are present in large quantities in the brain of teleost fish, have confirmed the presence of glial fibrillar acid protein (GFAP) [45,46] and Vim in them [44]. 

In the trout cerebellum, a relatively low level of vimentin (Vim) immunopositivity was revealed (Figure 7A), which is consistent with the data for optic tectum [3]. A few dense, intensely labeled Vim+ clusters of undifferentiated and elongated cells were found in the DMZ (Figure 7A, black inset, Table 4). The apical region contained a dense group of intensely labeled Vim+ undifferentiated cells (Figure 7A, dashed red oval). In the dorso-lateral part of the CC in the superficial region of the Ml, two types of intensely labeled Vim+ cells (Figure 7A, red arrow) were identified, forming a dense cluster (Figure 7A, black and red insets, Table 4). In the rostral dorso-lateral part of the CC, in addition to Vim+ cells, Vim+ granules were detected (Figure 7A, black inset, indicated by yellow arrowheads, Table 4). In the caudal part of the dorso-lateral region of the CC, four types of intensely labeled cells (Figure 7B, red arrows) were identified in the superficial and deep layers of the Ml (Figure 7B, black inset) with high or moderate immunolabeling intensity (Table 4). 

Vim+ granules in the caudal part of the dorso-lateral region of the CC were also identified in the Ml (Figure 7B, red inset). In the caudal dorso-lateral part of the CC, a moderate/high intensity of Vim immunolabeling was detected in pear-shaped cells of types 1 and 2 (Figure 7C, red arrows, black inset, Table 4). In the Gl of the same region, intensely labeled Vim+ elongated type 1 and 2 astrocyte-like cells were also found (Figure 7C, red inset, Table 4). In the lateral part of the CC, only intensely labeled Vim+ undifferentiated cells (Figure 7D, red arrows) and granules (Figure 7D, white arrows) were observed in the superficial region of the Ml (Figure 7D, black inset, Table 4). In the Grl of the central part of the CC, two types of elongated (Figure 7E, black inset, Table 4) and oval Vim+ cells (Figure 7E, red arrow, Table 4) with high/moderate labeling intensity were identified. 

In the GrEm, a heterogeneous population of Vim+ cells including five types was identified (Figure 7F,G, red arrows, Table 4). Rostrally, dense aggregations of Vim+ granules were found in the surface and inner layers of the GrEm (Figure 7F, red inset, black inset, white arrows). Single elongated and oval Vim+ astrocyte-like cells were localized in the deep layers of the GrEm (Figure 7F, red inset, red arrows). In the central part of the GrEm, a heterogeneous population of polygonal intensely/moderately labeled Vim+ cells (Figure 7G, red arrow) was present in the inner part (Figure 7G, red inset, Table 4) and dense aggregations of Vim+ granules were present in the surface layer (Figure 7G, black inset, white arrow). In the ventro-caudal areas of the GrEm, dense aggregations of Vim+ cells (Figure 7H, black inset, red arrows, Table 4) and Vim+ granules (Figure 7H, black inset, white arrow) were localized superficially. In the deep layers of the same GrEm region, single Vim+ cells with oval and polygonal morphology were observed (Figure 7H, red arrows). Rarely, single intensely labeled Vim+ cells were found in the surface layer of the GrEm (Figure 7I, black inset, red arrow).

A comparative study of the distribution of Vim+ cells in different areas of the cerebellum showed significant intergroup differences (*p* < 0.05) between the lateral and dorso-lateral parts of the CC, as well as between the GrEm and the Grl (Figure 7J). An estimation of the ratio of GS+ and Vim+ cells in the DMZ and in the lateral and dorso-lateral parts of the CC revealed significant intergroup differences (*p* < 0.01) in the DMZ and significant intergroup differences (*p* < 0.05) in the lateral and dorso-lateral regions (Figure 7K). 

### 2.8. Vimentin in the Trout Brainstem

In the brainstem, as in the trout telencephalon, a high level of Vim activity was detected (Figure 8A, Table 4). A heterogeneous population of Vim-negative (Vim−) cells (Figure 8A, yellow arrow), Vim+ cells (Figure 8A, red arrows), and Vim+ granules (Figure 8A, white arrows) with high or moderate intensity of labeling was identified in the lateral wall of the IV ventricle in the PVZ of the brainstem (Figure 8A, red inset, Table 4). 

At the bottom of the IV ventricle in the PVZ, undifferentiated intensely labeled Vim+ cells forming small clusters (Figure 8B, black inset, red dotted oval) and single elongated Vim+ cells (Figure 8B, black inset, red arrow) were found. The centrally located population of Vim+ cells at the bottom of the IV ventricle was represented by intensely labeled elongated cells forming small sparse groups (Figure 8B, red inset, red arrow) and dense clusters of undifferentiated cells (Figure 8B, red inset, red dotted oval). 

In the ventro-lateral part of the wall of the IV ventricle, undifferentiated intensely labeled Vim+ cells formed extensive areas (Figure 8C, black inset, red arrows); some areas included aggregations of intensely labeled Vim+ granules (Figure 8C, black inset, white arrows) and large dense groups of Vim− cells (Figure 8C, black inset, yellow dotted oval). At the bottom of the IV ventricle in this part of the brainstem, a generally similar distribution pattern was observed (Figure 8C, red inset). An examination of the cell composition of the PVZ at the bottom of the IV ventricle at a higher magnification showed the presence of a heterogeneous cellular composition with Vim positivity (Figure 8D, black inset, Table 4), which also included Vim+ RG cells (Figure 8D, black inset, red arrows).

Intense/moderate Vim immunopositivity was detected in large somatosensory interneurons (Figure 8E, red arrows) of the main (Figure 8E, black inset, Table 4) and lateral (Figure 8F, Table 4) nuclei of nerve V, the nucleus of the VII nerve (Figure 8G, Table 4), and in heterogeneous populations of the lateral (Figure 8H, black ovals, Table 4) and medial (Figure 8I, black inset, red arrows, Table 4) parts of the reticular formation. 

Large Vim+ cells were identified among RSCs (Figure 8J,K, red arrows, Table 4). A distinct population of elongated and undifferentiated intensely/moderately labeled Vim+ cells was found in the submarginal zone (SMZ) of the brainstem (Figure 8J,L, Table 4). Elongated Vim+ cells were identified immediately in the SMZ (Figure 8J, black inset, Table 4). In the superficial layers of the PVZ, single undifferentiated cells (Figure 8L, black inset, red arrows, Table 4) and their aggregations (Figure 8L, red dotted oval, Table 4), as well as Vim+ granules (Figure 8L, black inset, white arrow), were observed. The identification of Vim+ cells in the nuclei NIX-X of the parasympathetic dorsal column in the trout is consistent with data on rodents [35]. Large, polygonal, heterogeneous, intensely and moderately Vim+ labeled neurons were found in the trout NIX-X nuclei (Figure 8M, red frame, Table 4).

A comparative study of the distribution of Vim+ cells in different areas of the brainstem showed significant intergroup differences (*p* < 0.01) between the PVZ and RSC population, as well as significant intergroup differences (*p* < 0.05) between the groups of the PVZ and the medial reticular formation (MRF), the PVZ and the lateral reticular formation (LRF), and NVII and the RSC area (Figure 8N). The results of a two-way ANOVA of the proportions of Vim+, GS+, and DC+ cells in the PVZ, MRF, and LRF showed significant intergroup differences (*p* < 0.01) in the distribution of Vim and GS in the MRF and LRF and significant intergroup differences (*p* < 0.05) in the distribution of the same markers in the PVZ. In the distribution of GS and DC, significant intergroup differences (*p* < 0.05) were found in the MRF and LRF (Figure 8O).

### 2.9. Nestin in the Trout Cerebellum

Nestin (Nes) is a class VI intermediate filament protein expressed by neuronal progenitors in the subventricular zone (SVZ) of the lateral ventricle and the subgranular zone (SGZ) of the dentate gyrus in the adult brain [47]. Nes constitutes a part of the cytoskeleton, in particular intermediate filaments, and may have cytoplasmic localization. In differentiated cells, the Nes expression is replaced by the expression of proteins specific for neurons or glia. Nestin-positive (Nes+) cells can create neurons and glia during differentiation [48] and also influence the movement of vesicles in the cell, providing structural and functional support during cell proliferation [47,49]. Nes is involved in the control of localization and divergence of intermediate proteins in the cytoskeleton by acting on cellular components during mitosis [47]. The precursors of NE cells in neurogenic niches of the *D. rerio* cerebellum, subpallium, and periventricular zone (PVZ) of the tectum express Nes [50,51]. 

In the trout cerebellum, Nes+ cells were detected in the Ml, Gl, and in deeper parts of the Grl in the CC (Figure 9A, Table 5). Intense immunolabeling was detected in the dorsal part of the Ml and Gl (Figure 9A, black inset, Table 5). The Nes+ PC bodies were intensely/moderately labeled and represented by a heterogeneous population including five size types of PCs (Figure 9A, black inset, Table 5). In the periganglionic region, apical areas of Nes+ PC dendrites were identified, and a diffuse layer of intensely labeled Nes+ granules was detected in the basal part of the Ml (Figure 9B, black inset, Table 5).

A relatively homogeneous composition of Nes+ cells, including undifferentiated and elongated intensely labeled cells, was found in the DMZ (Figure 9F, red inset, Table 5). In the Ml near the DMZ, the distribution pattern of small undifferentiated Nes+ cells and RG cells was similar to that in the dorsal part of the CC (Figure 9F, yellow box). In the Gl of the dorsal part of the CC, clusters of oval, intensely labeled Nes+ cells were detected (Figure 9G, red dotted oval); clusters of undifferentiated Nes+ cells were found in the superficial region of the Ml, intensely labeled (Figure 9G, in the yellow rectangle and oval). Apical dendrites of Nes+ PCs penetrated the Ml and had a moderate intensity of labeling (Figure 9G,H, blue arrows). 

A heterogeneous population of Nes+ cells was identified in the GrEm (Figure 9I, Table 5). Small, undifferentiated, intensely labeled Nes+ cells were detected in the surface layers (Figure 9I, in the black box); single oval Nes+ cells were found in deeper layers (Figure 9I, red arrow); and diffuse aggregations of Nes+ cells were also found in deeper layers (Figure 9I, red box). Extensive aggregations of undifferentiated immunopositive cells (Figure 9J, red arrows) were frequently located in the submarginal zone (SMZ) (Figure 9J, black inset). A large aggregation of oval and elongated Nes+ cells with moderate or intense labeling was detected in the deep parenchymal zone of the GrEm (Figure 9K, red arrows). In the Ml basal zone, clusters of intensely labeled Nes+ cells were found in the PVZ (Figure 9L, black inset, red arrow) and SVZ (Figure 9L, white dotted inset, blue arrow).

A comparative study of the distribution of Nes+ cells in different areas of the cerebellum showed significant intergroup differences (*p* < 0.05) between the basal and lateral zones of the CC, as well as between the GrEm and the basal zone (BZ) (Figure 9M). An estimation of the ratio of Nes + and GS+ cells in the same areas of the CC and GrEm revealed significant intergroup differences (*p* < 0.05) in the GrEm and in the lateral and dorsal zones of the CC (Figure 9N).

### 2.10. Nestin in the Trout Brainstem

The immunolocalization of nestin (Nes) was detected in various regions of the trout brainstem, including the periventricular layer (PVL), NV, NVII, IX-X nuclei, dorsal reticular formation (DRF), reticulospinal cells (RSCs), and ventro-lateral external cells (VLCs) (Table 5). A general view of the distribution of Nes+ cells in the ventro-lateral part of the brainstem is shown in Figure 10A. In the PVL along the wall of the IV ventricle, three types of intensely labeled undifferentiated and elongated cells were identified (Figure 10A, blue inset, Table 5). 

In the ventro-medial region of the reticular formation, a heterogeneous population of Nes+ cells and a neuropil were found (Figure 10A, red rectangle). In the lateral external region, a population of intensely labeled, mostly small cells was observed (Figure 10A, blue box, Table 5). Several types of Nes+ cells were identified in the DRF (Figure 10B, Table 5). Small, intensely labeled, undifferentiated Nes+ cells were localized in the PVZ (Figure 10B, red inset, red arrows). Nes+ cell extended processes into the SVZ and deeper layers, forming a neuropil (Figure 10B, white inset, white arrow). In the caudal part of the DRF, numerous intensely/moderately labeled elongated, oval, or polygonal cells were distributed among the neuropil fibers (Figure 10C, white inset, Table 5). 

In the ventro-lateral external region of the trout brainstem, undifferentiated, elongated, and oval intensely labeled Nes+ cells were present (Figure 10D, black inset, red arrows, Table 5). At a higher magnification, PVL projections showed dense diffuse distribution patterns of Nes+ undifferentiated cells and radial glia (Figure 10E, black inset, red arrows). In the caudal regions of the brainstem, the patterns of distribution of immunopositive cells in the lateral part of the PVL were sparser (Figure 10F, black inset, red arrows), Nes+ neuropil fibers and heterogeneous groups of Nes+ cells were detected in the subventricular layer (SVL) of the lateral wall of the IV ventricle (Figure 10F, cyan box, red arrow), and dense clusters of undifferentiated Nes+ cells were localized in the ventral wall of the IV ventricle (Figure 10F, red box). 

In the rostro-lateral wall of the *fossa rhomboidea*, Nes+ undifferentiated cells were present in both the PVZ and SVL (Figure 10G, red inset). In the SVZ, larger oval and elongated, intensely/moderately labeled Nes+ cells were located under a layer of undifferentiated cells (Figure 10G, red arrows). In the caudal part of the *fossa rhomboidea* in the lateral wall, the patterns of distribution of undifferentiated intensely labeled Nes+ cells in the PVZ were less dense than in the rostral one (Figure 10H, black inset, blue arrows); no Nes+ cells were found in the SVZ. 

Within the main nucleus of the trigeminal nerve (nV), large, polygonally shaped, heterogeneous, intensely labeled Nes+ neurons were identified (Figure 10I, blue box, red arrows, Table 5). Three types of intensely/moderately labeled polygonal Nes+ neurons were found in the facial nerve nucleus (nVII) (Figure 10J, red square, blue arrows, Table 5). 

At different levels of the brainstem, single or small groups of polygonal-type, intensely/moderately labeled Nes+ RSCs were found (Figure 10K, black inset, red arrows, Table 5). Larger groups of oval or elongated, intensely/moderately labeled Nes+ RSCs, forming ventral (RSCv) and internal (RSCi) populations, were observed rostrally, in the region of the descending trigeminal tract (TrDV) (Figure 10L, in blue boxes, Table 5). The identification of cells in the NIX-X parasympathetic dorsal column in trout Nes+ cells is in line with rodent data [35]. Large, polygonal, heterogeneous, and intensely and moderately Nes+ labeled neurons were identified in the trout NIX-X nuclei (Figure 10M, red frame, Table 5).

A comparative study of the distribution of Nes+ cells in different areas of the brainstem showed the presence of multiple significant intergroup differences (*p* < 0.01) between the dorsal reticular formation (DFR) and RSC population, PVZ and SVZ, nV, nVII, and also significant intergroup differences (*p* < 0.05) between the VLC and RSC populations (Figure 10M). An estimation of the ratio of Nes+ and DC+ cells in several areas of the brainstem revealed significant intergroup differences (*p* < 0.05) in the DRF and the VLC population (Figure 10N).

### 2.11. Double Immunolabeling in the Brainstem and Cerebellum of Trout

We conducted a series of experiments for the identification of double immunolabeling to elucidate the phenotypes of proliferating cells in the caudal regions of the trout brain. The immunolabeling of GS and PCNA revealed the colocalization of these markers in the NECs located in the PVZ of the brainstem (Figure 11A). In deeper parenchymal layers, GS+ RG fibers (Figure 11B), as well as monolabeled GS+ and PCNA+ cells (Figure 11B), were identified. In some cases, patterns of GS/PCNA colocalization in the type 1 cells were identified (Figure 11B). In the parenchyma of the brainstem, patterns of colocalization of GS/PCNA in individual cells were also detected (Figure 11C), but mono-immunolabeled GS+ and PCNA+ cells have prevailed (Figure 11C). While investigating the localization patterns of Vim and PCNA in the PVZ, the cells with double labeling were identified (Figure 11D). Monolabeled Nes+ and PCNA+ cells, as well as cells with Nes+/PCNA+ colocalization, were identified in the CMZ of the brainstem (Figure 11E). In some cases, local constitutive neurogenic niches (CNNs) containing dense clusters of Nes+ cells were found in the SMZ (Figure 11F). The study in the cerebellum did not reveal the cells with double immunolabeling (Figure 11G); nevertheless, the patterns of distribution of Nes+ cell clusters in the surface zone of the ML in combination with GS+ RG fibers were revealed (Figure 11H). In some cases, discrete diffuse clusters of GS+ and Nes+ cells in the cerebellum have been identified (Figure 11F). 

## 3. Discussion

In the present study, we, for the first time, carried out a comparative analysis of cellular proliferation and the distribution of GS, Vim, and Nes markers, as well as DC, in the cerebellum and brainstem of 3-year-old rainbow trout, which revealed aNSPCs. The results obtained made it possible to identify the neurons formed in the adult period of life in this fish species. Unlike the forebrain regions of the brain (telencephalon and tectum) [3], the caudal ones (cerebellum and brainstem) have not previously been considered in adult salmon. The present study focused on the distribution of these molecular markers underlying adult neurogenesis in the context of the putative multipotency in aNSPCs. We were guided by the hypothesis of potential heterogeneity of aNSPCs in the caudal regions of the brain and their different ways of molecular control, which represents a novel strategy in this area of research. In fish, aNSPCs persist throughout life in various niches of stem cells in the brain and spinal cord [12]. Populations of stem cells, including NECs and RGCs, develop throughout life in neurogenic areas of the brain. 

Within the brain, RG cell domains remain in a cyclic or quiescent state, while NE cells are in a state of continuous division [24]. Taking into consideration that most PCNA-labeled cells in the periventricular and outer submarginal zones of the trout brainstem are NE cells, it is reasonable to assume that they continuously proliferate in the trout brainstem and produce the major part of GS+, Vim+, and Nes+ cells. Considering that most PCNA-labeled cells in the periventricular and outer submarginal zones of the trout brainstem are NE cells, it is reasonable to assume that they continuously proliferate in the trout brainstem and produce most of the GS+, Vim+, and Nes+ cells. The results of double immunolabeling have shown the presence of double immunolabeling patterns of GS+/PCNA+ and Vim+/PCNA+ in the PVZ of the trout brainstem. This indicates that some of the GS+ cells can be attributed to a population of slowly proliferating precursors. The identification of the double labeling of GS+/PCNA+ and Vim+/PCNA+ in single cells of the parenchyma of the brainstem probably shows proliferative activity in glial cells, which corresponds to data on zebrafish [1,18] and *N. furzeri* [21]. 

The detection of the colocalization patterns of Nes+/PCNA+ in the SMZ of the brainstem of trout is an interesting finding, indicating the presence of adult constitutive neurogenesis in this area, which is not typical for other fish species. The colocalization of the studied markers, however, is characteristic of a limited population of the PVZ and PZ cells in the brainstem of trout; thus, in the intact state, homeostatic neurogenesis in adult trout is limited. The study of the distribution patterns of Nes and GS in the cerebellum of trout did not reveal areas of double labeling, which obviously determines the species-specific features of the NSPC phenotypes. 

Heterogeneous RGC populations were also found in the region of the central canal of the spinal cord, showing episodic proliferative activity under homeostasis conditions [12]. A recent study of adult neurogenesis in *D. rerio* and CNS neuroregeneration has shown that the proliferation of aNSPCs is regulated by a variety of stimuli that may be related to biological functions. In *Nothobranchius furzeri*, the DC expression markedly decreases with age, as in many mammalian species studied to date [52,53]. The decrease in mitotic activity coincides in time with the decrease in DC labeling [52]. A growing body of evidence suggests that aNSPCs are sensitive to endocrine signals [54], specific sensory modulation [55], social interactions [56], nutrient availability [57], and neurotrauma [8,9,13,20,22,58]. Depending on these factors, different populations of stem cells and progenitor cells can change their cycles of activity. Such stimuli appear to act as triggers that either turn on normally dormant aNSPCs or modulate the homeostatic cycle rate and behavior of niche-specific cells. The identification of various forms of stimuli that affect RGC and NEC proliferation, along with the identification of key molecular regulators, contributes to our understanding of the relationship between the aNSPCs’ activity and their biological significance. 

### 3.1. Adult Cerebellar Neurogenesis and Expression of Molecular Markers by Various Types of aNSPCs

The activity of cerebellar progenitors and neurogenesis continue during the adult stage in various fish species [52,59,60]. Progenitor cell activity was observed along the IV ventricle, as well as in the dorso-medial and lateral regions of the CC and the cerebellar recess [24]. In the cerebellum of adult zebrafish, *D. rerio*, progenitors are stored in specialized niches [30]. In studies on adult *D. rerio*, GFAP+, BLBP+, S100b+, and Vim+ RG cells were identified in the region of the cerebellum midline, in the molecular layer next to the DMZ where granular neurons are formed [50]. In case of damage to the pallium, RG upregulates the expression of intermediate filament proteins such as GFAP, Vim, Nes, and calcium-binding protein S100b and exhibits hypertrophy. These structural changes are reminiscent of reactive gliosis in mammals, but in teleost fish, the scar formation processes are weakly expressed [61].

Proliferating cells are mapped in this study by PCNA immunohistochemistry, which has been superseded over the course of two decades by the detection of another cyclic cell marker, Ki-67, since PCNA is related to a DNA repair mechanism and thus may give false positive labels for non-cycling cells. However, PCNA immunohistochemistry can still be used for the in situ proliferation assays if false positive results can be excluded, which is the case in the present study for two objective reasons: (i) the proliferation of neural progenitors/stem cells has been repeatedly described in fetal and adult fish maps consistent with those currently observed [59,60], and (ii) the PCNA-immunopositive cells in the present study show discrete localizations in contrast to the regularly sparse PCNA-immunonegative cells.

It has been shown that in the cerebellum of brown ghost knifefish, *A. leptorhynchus*, approximately half of Sox2+ cells (a marker of stem cells) in the DMZ also express GFAP and Vim, whereas a respective proportion of Sox2+ cells are positive for the astrocyte S100 protein. The large number of such cells indicates that the populations of S100+/Sox2+, GFAP+/Sox2+, and Vim+/Sox2+ cells are either identical or similar in many respects. Based on these data, Sîrbulescu and co-authors [6] suggest the presence of a homogeneous population of glial progenitor cells in *A. leptorhynchus*.

The constitutive neurogenic niches in the *A. leptorhynchus* cerebellum consist of non-astrocytic cells, presumably epithelial/ependymal cells, a large proportion of glial cells expressing GFAP, S100, Vim, and a small proportion of BLBP [6]. This assumption is further supported by the high degree of similarity between the expression domains of these glial markers, suggesting a single, homogeneous population of glial progenitor cells. However, studies on *D. rerio* have shown that the DMZ is not labeled with canonical radial or astroglial markers such as Vim, GFAP, and BLBP but contains Nes and other stem cell markers, thus indicating that these cells retain neuroepithelial rather than astroglial characteristics. Kaslin suggests [30] that these highly proliferating Nes+ cells of epithelial origin are the main/only source of neurons in the granular layer of adult *D. rerio*. Apparently, in *A. leptorhynchus*, the differences between the ventricular zone and progenitor cells of the upper rhombic lip are not as pronounced as in *D. rerio*.

In rainbow trout, undifferentiated cells expressing GS, Vim, and Nes were found in the matrix zones of the cerebellum: the DMZ, lateral and medial surfaces of the roof of the IV ventricle, superficial dorso-medial and dorso-lateral regions of Ml, and superficial regions of the GrEm. In most of these zones, PCNA+ cells were also detected. These cells differed in molecular profile, localization, size characteristics, and pattern of grouping into clusters, which probably indicates the heterogeneity of such populations in the trout cerebellum. Single PCNA+ radial glial fibers are not routine observations by PCNA immunohistochemistry. PCNA is indeed a nuclear antigen, and thus specific PCNA immunoreactivity typically consists of labeled nuclei. RG in the vertebrate brain are an exception to this rule, as their radial fibers do express PCNA immunoreactivity. This was previously shown in another article on post-traumatic reactions in the adult trout brain [36].

Thus, both the cells with NE morphology and the RG cells were isolated among GS+ precursors in trout (Figure 3C,D,G). Vim+ cell precursors had an exclusively NE morphology and formed extensive clusters predominantly in the superficial layers of the cerebellum, DMZ, and the GrEm, where they were also present in the parenchymal part (Figure 7G). Nes+ cells included both NE and RG cells located in the DMZ (Figure 9F), deep in the Ml (Figure 9C), and forming small immunopositive clusters in the basal part of the Ml (Figure 9G). So, the populations of aNSPCs in the trout cerebellum expressing GS, Vim, and Nes have different morphologies and differ in localization and the pattern of cluster formation, which indicates the morphological and, obviously, functional heterogeneity of these cells.

In zebrafish, NE-like cells were distinctly polarized and expressed markers of neural stem cells and nestin precursors, Sox2, Meis, and Musashi1, located along the IV ventricle and its dorsomedial and lateral parts, the cerebellar depression [50,62]. It is known that Nes-positive progenitors in this fish species do not express radial or astroglial markers such as S100b, GFAP, Vim, BLBP, GS, or Aro-B [50]. In rainbow trout, immunopositivity to Nes in the cerebellum is significantly higher than to Vim; in addition, Vim+ cells without processes were found in Grl, but Nes+ cells were not observed. The presence of Nes+ RG cells in the cerebellum distinguishes rainbow trout from zebrafish, whose cerebellum is reported to lack Nes+ RG. 

Furthermore, the presence of Vim+ NE-type cells and the absence of Vim+ RG in trout also indicate significant differences between trout and zebrafish aNSPCs. In zebrafish, the cerebellar niche of stem cells consists of polarized neuroepithelial-like progenitors located in the dorsal part of the cerebellar recess, while the S100b- and GFAP-expressing epithelial-like glial cell line occupies the ventral part of the IV ventricle [50,62]. In addition, neuroepithelial-like progenitors border medially on RG and laterally on Bergmann glia. The trout DMZ contains NE-type cells labeled with GS, Vim, and Nes (Figure 3C,D, Figure 7A and Figure 9F). 

The morphologies of NE cells and their aggregations, as well as their localizations within the DMZ, differed markedly in rainbow trout. In particular, GS+ NE cells were of two types: with a typical elongated morphology and with a rounded morphology (Figure 3C,D). Cells of both types were evenly distributed in the lateral walls of the DMZ. Vim+ aNSPCs were represented only by small rounded undifferentiated cells and were localized in the medial part of the DMZ (Figure 7A). Rounded Nes+ cells formed dense clusters in the ventral part of the DMZ (Figure 9F). Thus, the morpho-topographic characteristics of GS+, Vim+, and Nes+ cells in the DMZ differed, which indicates a heterogeneous pattern of progenitor populations in trout. With DC labeling in the cerebellum, the largest number of DC+ neuroblasts were detected in the DMZ, where their number significantly (*p* < 0.01) exceeded that in the Grl (Figure 5Q).

In zebrafish, neuroepithelial-like progenitors give rise to a population of dividing intermediate progenitors [62,63]. Neuroepithelial-like cells and intermediate progenitors express markers characteristic of the rhombic lip and granule cells such as atoh1c, zic1, zic3, rielin, neurod, Pax6, Meis1, and GAP-43 [50,62,63]. The intermediate progenitors quickly migrate outside into the granular cell layer, where they differentiate into granule cells. In the adult zebrafish cerebellum, the granule cell precursors migrate into the granule cell layer within 3 days [50,62]. 

In rainbow trout, we detected a few Vim+ cells in the Grl of the CC (Figure 7E), Vim+ and Nes+ cells in the parenchymal part of the GrEm (Figure 9F,G), and aggregations of DC+ neuroblasts in the basal part of the Ml (Figure 5I). Vim+ and Nes+ cells detected in the GrEm were significantly larger than those in the DMZ (Table 4 and Table 5), which allows us to consider them as precursors of granular cells. In the same area, DC+ cells of a heterogeneous morphology were identified, which indicates a long process of constitutive formation of new neurons in the GrEm. The presence of intensely labeled DC+ projection cells (PCs and EDCs) in the dorsal (Figure 5H), lateral (Figure 7J,K), and basal (Figure 5B,C,E) parts of the CC suggests that the formation of long-axon extracerebellar projections in the trout cerebellum continues throughout a life span.

Although several subtypes of inhibitory and excitatory cells have been found in the zebrafish cerebellum, granular cells are predominantly formed in adults [50]. In trout, we also found a large number of DC+ cells in the Grl (Figure 5L,M). The appearance of re-induced DC in projection cells and granular neurons of various types in the trout cerebellum is associated with the high plasticity of these cells. An estimation of the ratio between the number of DC+ cells and GS+ RG showed that the density of GS+ RG fibers significantly exceeded the number of DC+ neuroblasts in the dorsal and lateral parts of the CC, while no significant differences were found in the basal zone and the DMZ (Figure 5R). It should be noted that the ratio of the numbers of DC+ and GS+ cells did not differ significantly in all areas of the cerebellum (Figure 5S).

In vivo imaging of the cerebellar stem cell niche from embryonic to adult stages has shown that the niche is created in a two-step process, initially involving morphogenetic cell movements and then tissue growth [24,50,64]. Through these processes, the rhombic lip precursors and a small part of the ventricle, the cerebellar recess, are displaced deep into the cerebellar tissue [24,50]. In chickens and mice, in contrast, the cerebellar midline and the ventricle are lost as the two cerebellar hemispheres fuse [65,66]. 

The progenitors of the upper rhombic lip persist dorsally to the cerebellar recess in adult zebrafish, while progenitors derived from the ventral zone (VZ) and glia are located ventrally in the IV ventricle [24,50]. VZ-derived progenitors are almost inactive in adult zebrafish, which is consistent with negligible formation of inhibitory neurons and glia during the adult stage. In trout, GS+ and Nes+ cells were found in region IV (Figure 3K,L and Figure 9L). The diffusely located single GS-immunopositive cells were present in the lateral parts of the dorsal roof of the IV ventricle (Figure 3K) and grouped into small clusters of GS+ type NE cells in the medial ones (Figure 3L), which distinguishes rainbow trout from other fish species, in particular, zebrafish. In the medial part of the dorsal roof of the IV ventricle, significant immunopositivity to nestin was also revealed (Figure 9L), thus indicating a significant neurogenic potential of the trout VZ.

During the ontogeny in amniotes, granular cell precursors migrate from the rhombic lip to the surface of the cerebellum, where the highly proliferative secondary germinal zone (SGZ) is temporarily formed [66]. In rainbow trout, superficial Vim+ (Figure 7A,B,H) and DC+ (Figure 5G,P) cell populations were found in the dorsal and lateral zones of the corpus cerebellum (CC) and the lateral zone of the GrEm. The emergence of the secondary proliferative zone (e.g., the outer layer of granular cells) seems to be a specific developmental adaptation of amniotes because sharks and zebrafish lack the pronounced outer granular cell layer [67]. 

In rainbow trout, in the basal part of the cerebellar body, there is a median Ml invagination with a large number of DC+ neuroblasts of the EDC projection located along it (Figure 5D,F). In addition, large DC+ cells in the granular layer resemble ectopic projection cells more than granular cells (Figure 5K,L). The division of intermediate granular progenitors with a low proliferation rate is found in the zebrafish cerebellum [51]. This suggests that the formation of granular cells in zebrafish is most likely controlled at the level of primary progenitors [51]. 

Thus, in adult rainbow trout, the cerebellum retains active processes of homeostatic neurogenesis and has several proliferative neurogenic zones containing aNSPCs. The cell composition and molecular profile of progenitors in different neurogenic areas of the cerebellum vary, which confirms the heterogeneous pattern of neurogenic niches in the adult rainbow trout cerebellum. Although several subtypes of inhibitory and excitatory cells have been found in the zebrafish cerebellum, granular cells are predominantly formed in adults [39]. In rainbow trout, we also found a large number of DC+ cells in the Grl (Figure 5L,M). 

An estimation of the ratio between the number of DC+ cells and GS+ RG showed that the density of GS+ RG fibers was significantly higher than the number of DC+ neuroblasts in the dorsal and lateral parts of the CC, while no significant differences were found in the basal zone and the DMZ (Figure 5R). It should be noted that the ratio of the numbers of DC+ and GS+ cells in all areas of the cerebellum also showed no significant differences (Figure 5S).

### 3.2. Adult Neurogenesis in the Brainstem

Unlike the forebrain, the caudal regions of the brain, in particular, the brainstem located caudally to the isthmus, remain the least studied structures. Teleostei comprise more than 26,000 fish species with a wide variety of habitats, a wide range of morphological adaptations, and varying growth rates and sizes of different brain centers [18]. The regeneration of motor neurons after a spinal cord injury has been reported for adult zebrafish [68]. The specific environmental conditions force *N. furzeri* to have a very rapid development to adult size [52,69], including accelerated brain growth and neurogenesis. Recent studies suggest that this can be explained by the long-term retention to adulthood of a highly neurogenic embryonic line rather than by the increased performance of existing bloodlines [21]. This study has shown a diversity of neurogenic adaptations in the adult fish brain and illustrated the ability of fish models to utilize various natural strategies that can be used to enhance neurogenesis.

Currently, increasingly more attention is being paid to the implication of sensory stimuli for modulating the processes of adult neurogenesis, including the brainstem [12]. However, the structural features of the respective regions of the brain, as well as their relationship with the reticular formation of the brain, remain unclear. In this regard, we consider the GS, Vim, and Nes expression in the cells of the neurogenic zones with PCNA+ activity of the trout brainstem, as well as the presence of DC+ cells in the reticular formation and its derivatives and sensory nuclei of V and VII cranial nerves, as well as the parasympathetic column of the IX-X cranial nerves of the brain, in the context of the heterogeneity of aNSPCs and also the specific influence of somato- and viscerosensory signaling on the development of medulla oblongata centers that process these types of sensory signaling in rainbow trout. 

In the trout brainstem, which contains projections of the cranial nerve nuclei, in the lateral, medial, and dorsal parts of the reticular formation, the expression of GS, Vim, Nes, and DC was found in various cell populations (Table 2, Table 3, Table 4 and Table 5). Of particular interest are the expression patterns of Vim, Nes, and DC in the parasympathetic nuclei of the IX-X cranial nerves of the trout. This issue is particularly important since the neurogenic niche has been described in detail in the vagal complex of the adult rodent brain [46]. The expression of Nes and Vim is associated with a positive regulation of the aNSPCs’ proliferative activity. In the PVZ of the isthmic part of the brainstem, as well as in the submarginal areas of the cranial nerves projections, the dorsal reticular formation, a high number of PCNA+ cells (Figure 2A–D) and local aggregations of PCNA-negative cells in the SVZ were found (Figure 2E,F). The projections of the cranial nerve V, VII, and IX-X pairs are the somato- and viscerosensory nuclei located in the dorsal column of the medulla oblongata and processing sensory information from the sensory receptors of the head (the nuclei of the V and VII pairs of nerves) and visceral organs (the IX-X pairs of nerve nuclei). Sensory input is considered a powerful regulator of animal behavior [12]. 

Teleost fish, being inhabitants of the aquatic environment, receive this information from visual cues, olfactory and gustatory cues (e.g., chemosensory), and lateral line input (i.e., mechano-sensory). In rainbow trout, numerous GS+ RG fibers were found in the brainstem, directed both from the primary PVZ near the IV ventricle (Figure 4A–C) and from the outer submarginal zone (SMZ) of the brain (Figure 4F,I–L). A high distribution density of GS+ RG was also revealed in the interfascicular zone (IFZ, Figure 4B,F,G) and the region of the descending tract of the V nerve (ITZ, Figure 4E) of trout, which clearly indicates the involvement of RG fibers in the migration of adult cells to the IFZ and the main nucleus of the V nerve. 

A one-way ANOVA showed the largest number of trout GS+ cells in the PVZ, IFZ, and the ventro-lateral region of the reticular formation, which significantly differed (*p* < 0.05) from the number of cells in the ITZ (Figure 4M). The results of a comparative analysis showed that undifferentiated GS+ cells numerically dominated all the brainstem areas compared to RG cells (Figure 4N), but no significant differences were found.

PCNA+, GS+, Vim+, and Nes+ cells of the undifferentiated type, representing aNSPCs located in neurogenic niches, were also identified in all these areas (Table 1, Table 2, Table 4 and Table 5). The results of the two-way ANOVA showed a significant dominance (*p* < 0.05) of GS+ aNSPCs compared to Vim+ aNSPCs in the SGZ and a significant dominance (*p* < 0.01) in the MRF and LRF (Figure 8O). We suggest that the primary PVZ and secondary SMZ of neurogenic areas of the trout brainstem contain a pool of continuously forming progenitors, which also confirms the presence of DC+ cells in these areas (Figure 6A,M,N). The results of a comparative analysis of the ratio of DC+ cells and GS+ undifferentiated aNSPCs and GS+ RG showed approximately the same numbers of DC+ cells and GS+ RG in the PVZ, IFZ, and VLC/SMZ (Figure 6R). However, the ANOVA data indicate a significant dominance (*p* < 0.05) of GS+ aNSPCs compared to DC+ cells in the IFZ and VLC/SMZ (Figure 6S) and a significant dominance (*p* < 0.01) of the total number of GS+ RG and cells compared to DC+ cells in the same areas of the trout brainstem (Figure 6T).

As has been shown in zebrafish studies, aNSPCs exist in neurogenic niches, participating in the primary processing of sensory structures in the mature brain, including the olfactory bulbs (smell) in the forebrain, optic nerves in the midbrain tectum (vision), and vagus/facial hindbrain lobes (taste) [12,70]. These sensory domains are of interest for studying a range of modality-specific cues and their effect on the aNSPCs’ lineage activity. In rainbow trout, the somatosensory nucleus (VII pair of nerves) and viscerosensory nuclei (IX–X pairs of nerves) contained numerous DC+ cells, which indicates their postembryonic homeostatic generation in trout (Figure 6B,C,G). 

The results of ANOVA showed that the maximum number of DC+ cells in adult rainbow trout is localized in the primary proliferative zone of the PVZ and significantly exceeds (*p* < 0.05) that in the secondary SMZ (Figure 6Q). Thus, in the adult state, the PVZ in the medulla continues to produce a greater number of new cells compared to the submarginal zone. Other species such as zebrafish, medaka, brown ghost knifefish, goldfish, and killifish [12,67] have been also shown to have similar sensory niches. Thus, these sensory projection regions in the brain constitute an area for wide comparison of the functional role of aNSPCs using species-specific biologically appropriate forms of sensory stimuli. 

The constant renewal of cells in various zones of the reticular formation plays a special role in the homeostatic growth of rainbow trout. A large number of DC-immunopositive cells were detected in the interfascicular zone (IFZ) (Figure 6D,E), the dorsal reticular formation (DRF) (Figure 8H), and the population of reticulospinal cells (RSCs, Figure 6L,M). A comparative analysis of the number of the adult-born DC+ cells in the dorsal and ventro-lateral reticular formation showed a significant dominance of DC+ cells in the DRF (*p* < 0.05) compared to the population of RSCs (Figure 6Q). Morphologically, this conclusion does not seem obvious, especially taking into account the large number of DC+ cells in the ventro-lateral reticular formation. As the trout body size increases, there is a proportional increase in the somatosensory and viscerosensory projection areas in the brainstem. Thus, the increase in the cell number and emergence of new cells, in particular RSCs, in the reticular formation of adult animal naturally reflects a change in the cell composition of neural networks innervating the increasing surface of the animal’s body. 

Studies using forms of environmental enrichment or selective visual cues have shown a strong relationship between sensory input and the aNSPCs’ activity [12]. A recent study has provided a clear example of how simply the possibility of visual stimuli can induce long-term changes in the aNSPCs’ activity [56]. In this work, socially acclimatized zebrafish were isolated before being exposed to conspecifics in an adjacent tank chamber, which was sufficient to enhance the forebrain aNSPCs’ activity. This conclusion is supported by an early study where zebrafish were exposed to enriched environments adorned with aquatic plants and gravel or devoid of such elements, demonstrating an overall increase in the forebrain aNSPC proliferation with the enrichment [69,71]. 

Along with undifferentiated NE cells and RG, Vim+ and Nes+ cells were also found in the cranial nerve V, VII, and IX-X nuclei in rainbow trout (Figure 8E–G,M and Figure 10I,J,M). Vimentin and nestin are intermediate filament proteins that are part of the intracellular skeleton and can be defined as a structural component of the cell [44,72]. The same feature was characteristic of large reticulospinal cells in trout (Figure 8I–K and Figure 10K,L). The ANOVA results, however, showed a significant dominance of Vim+ aNSPCs in the PVZ over Vim+ RSCs (*p* < 0.01) and a significant dominance (*p* < 0.05) over Vim+ MRF and LRF cells (Figure 8N). Comparative intergroup studies of the number of Nes+ aNSPCs in the PVZ showed a similar significant dominance of such cells (*p* < 0.01) over Nes+ SVZ cells, as well as projections of the V and VII nerves (Figure 10N). The results of ANOVA of Nes+ cells of the reticular formation revealed a significant dominance of Nes+ cells in the DRF over RSCs (*p* < 0.01) and a significant dominance (*p* < 0.05) of SMZ/VLC over RSCs (Figure 10N).

Thus, different types of aNSPCs, GS+ RG, and NE cells were identified in the trout brainstem in the PVZ, SMZ, and IFZ. Additionally, GS+ RG bundles were detected in the region of the descending tract of the V nerve and IFZ. With PCNA+, Vim, and Nes immunolabeling, NE-type aNSPCs were detected in the PVZ and SMZ. However, comparative studies of the distribution of Nes+ and DC+ cells showed that the number of such cells in the primary proliferative zone does not vary significantly, while the number of Nes+ aNSPCs in the SMZ and DRF significantly (*p* < 0.05) exceeds the number of DC+ cells (Figure 10O). 

Taking into account that the numbers of GS+ and Nes+ aNSPCs in the PVZ are approximately the same and correspond to the number of DC+ cells, while the number of Vim+ cells is significantly (*p* < 0.05) lower than the number of DC+ cells, we suggest that GS+ and Nes+ cells dominate the over the primary PVZ of the adult trout brainstem. Vim+ aNSPCs of the NE type are much less represented. In the SMZ, the situation is different, as we observed almost no GS+ and Vim+ aNSPCs of the NE type, but we detected GS+ RG and Nes+ aNSPCs. Vim+ and Nes+ cells with the RG phenotype were not detected in the PVZ and SMZ of the trout brainstem. Heterogeneous populations of the Nes+ cells were found in the IFZ and projection areas of the V and IX-X nerves. Heterogeneous populations of the Vim+ cells were found in the LRF, RSC, and projection regions of the V, VII, and IX-X nerves. In further studies, we hope to elucidate the functional role of these cell populations in the rainbow trout brainstem.

## 4. Material and Methods

### 4.1. Experimental Animals

This study was carried out on 20 male rainbow trout, *Oncorhynchus mykiss*, at an age of 3 years, with a body length of 28–35 cm and a weight of 290–325 g. A diagram of the caudal brain regions, including the cerebellum and the brainstem in the sagittal projection, is shown in Figure 11. The animals were obtained from the Ryazanovka experimental fish hatchery in 2021. The fish were kept in a tank with aerated fresh water at a temperature of 14–15 °C and fed once a day. The daily light/dark cycle was 14/10 h. The concentration of dissolved oxygen in the water was 7–10 mg/dm^3^, which corresponds to normal saturation. All experimental manipulations with animals were conducted in accordance with the rules listed in the charter of the A.V. Zhirmunsky National Scientific Center of Marine Biology (NSCMB) FEB RAS and the Ethical Commission regulating the humane treatment of experimental animals (approval # 1-100424 from Meeting No. 4 of the Commission on the biomedical ethics of NSCMB FEB RAS, 10 April 2024). Figure 12 shows the diagram of the trout brain, demonstrating the levels of the frontal sections through the cerebellum and brainstem (a–l).

### 4.2. Preparation of Material for Immunohistochemical Studies

#### 4.2.1. Anesthesia and Prefixation

The animals were removed from the experiment and euthanized by the method of rapid decapitation. The fish were anesthetized in a 0.1% solution of ethyl-3-aminobenzoate methanesulfonate (MS222) (Sigma, St. Louis, catalog no. WXBC9102V MO, USA) for 10–15 min. After the anesthesia, the intracranial cavity of the immobilized animals was perfused with a 4% paraformaldehyde solution prepared in 0.1 M phosphate buffer (pH 7.2). After the prefixation, the brain was extracted from the cranial cavity and fixed in a 4% paraformaldehyde solution for 2 h at 4 °C. The fixed brains were kept in a 30% sucrose solution at 4 °C for two days (with the solution changed seven times). Serial frontal brain 50 μm thick sections were cut on a freezing microtome (Cryo-star HM 560 MV, Oberkochen, Germany). Every third frontal section of the cerebellum and brainstem was taken for the reaction.

#### 4.2.2. Control and Specificity of Antibodies

Monoclonal antibodies against proliferating nuclear antigen (PCNA) have been used previously in our laboratory to label progenitor cells in the trout brain under intact conditions and as a result of traumatic injury to the trout eye [36]. PCNA is present in proliferating cells throughout the cell cycle, although its expression is stronger during the S phase [36]. Monoclonal antibodies against GS have previously been used as a marker for neuroepithelial and radial glial neuronal progenitors in the *Oncorhynchus masou* cerebellum [13] and in the telencephalon of juvenile chum salmon, *Oncorhynchus keta* [22]. Antibodies against DC have previously been used to characterize the immature neurons in the telencephalon and tectum of trout and chum salmon *O. keta* [3,8]. Monoclonal antibodies against vimentin were previously used in our laboratory to detect NSPCs in the mesencephalic tegmentum of juvenile chum salmon, *O. keta*, [8] and cerebellum of juvenile masu salmon, *O. masou* [13]. Monoclonal antibodies against nestin were used to detect NSPCs in the mesencephalic tegmentum of juvenile chum salmon, *O. keta*, [8] and the cerebellum of juvenile masu salmon, *O. masou* [13]. 

The specificity of all antibodies against molecular markers used in this work was tested by Western blotting of the trout cerebellum and brainstem proteins. The cerebellum and brainstem extracts were obtained from adult trout by standard methods (for further information on methods, see [73]). As a positive control, protein extracts of the brain of adult mice were used in parallel to assess the antibodies’ specificity. The loading and blotting of the same number of total proteins were verified using a membrane with a monoclonal mouse anti-β-actin antibody (1:5000; Sigma-Aldrich, Sent-Luis, MO, USA). The molecular weight of PCNA was compared with prestained molecular weight markers (Sigma). For PCNA, the manufacturer says that the antibody labels a band of about 34 kDa in Western blots in the mouse brain protein extracts. In the band of trout protein extracts, a single band of about 34 kDa can be recognized (Figure 13). The molecular weight of GS was compared to prestained molecular weight markers (Sigma); this corresponded to the 37 kDa isoform of the protein (Figure 13). According to the manufacturer, anti-GS antibodies are specific to *D. rerio*. Doublecortin is expressed in immature neurons and is also considered as a marker of migrating neurons. In the bands of trout protein extracts, it recognizes one band at about 43 kDa (Figure 13). Additionally, for doublecortin, a negative control of histological staining of the cerebellum and brainstem was performed in the absence of primary antibodies (Appendix A). Nestin is a type VI intermediate filament, constituting a part of the cytoskeleton, and may have cytoplasmic localization in differentiated cells. Nestin-positive (Nes+) cells can create neurons and glia during differentiation [48] and also influence the movement of vesicles in the cell, providing structural and functional support during cell proliferation [47,49]. The precursors of NE cells in neurogenic niches of the *D. rerio* cerebellum and periventricular zone of the tectum express Nes [50,51]. The molecular weight of Nes was compared with prestained molecular weight markers (Sigma); this corresponded to 50 kDa (Figure 13). Vimentin is a type III intermediate filament protein expressed by astroglial cells [43]. It is commonly regarded and used as a marker of astrocytic glia in the vertebrate brain [44]. Immunocytochemical studies of ependymal cells and radial glia, which are present in large quantities in the brain of teleost fish, have confirmed the presence of Vim in them [44]. The molecular weight of Vim was compared with prestained molecular weight markers (Sigma); in the trout protein extract bands, Vim recognizes one band at about 57 kDa (Figure 13). 

#### 4.2.3. Western Immunoblotting 

For Western blotting, 10 intact adult rainbow trout were used. The animal brain was removed from the skull in 0.01 M Tris–HCl buffer (pH 7.2). The brain samples were rapidly cooled and homogenized in three times the volume of ice-cold buffer on a Potter–Elvehjem polytetrafluoroethylene glass homogenizer (Sigma). Homogenization buffer contained 20 mM Tris–HCl buffer (pH 7.2) supplemented with 0.25 M sucrose, 10 mM EGTA, 2 mM EDTA, and protease inhibitors: 2 mM PMSF, 50 mg/mL leupeptin, 25 mg/mL aprotinin, 10 mg/mL pepstatin, and 2 mM dithiothreitol. Homogenates of the cerebellum and brainstem of the rainbow trout were centrifuged for 15 min at 15,000× *g* on a Beckman Coulter Ti50 Rotor. A 50 mg aliquot of the homogenate was applied to the protein lane and separated using sodium dodecyl sulfate in resis polyacrylamide gel (SDS PAGE) on a 10% polyacrylamide gel. After electrophoresis, the isolated protein was accurately transferred to a nitrocellulose membrane and left overnight in 0.01 M Tris–HCl buffer (pH 8.0) supplemented with a 0.15 M NaCl solution containing 4% BSA (Sigma). The membranes were washed with distilled water and incubated with mouse monoclonal antibodies against GS (Catalog # ab64613, Abcam, Cambridge, UK; diluted 1:100), PCNA (Catalog # Sc-53407, Santa Cruz Biotechnology, Santa Cruz, CA, USA; diluted 1:100), doublecortin (Catalog # CO613 sc-271390, Santa Cruz Biotechnology, CA, USA, diluted 1:100), vimentin (Catalog # 3B4 ab28028, Abcam, Cambridge, UK; diluted 1:100), and nestin (Catalog # 2C1.3A11 ab18102, Abcam, Cambridge, UK, diluted 1:100) in 0.01 Buffer M Tris–HCl containing 1% BSA and 0.2% Tween-20 for 3 h at room temperature. The membranes were then washed with shaking in the 0.01 M Tris–HCl buffer containing 0.2% Tween-20 and incubated with horse anti-mouse secondary antibodies (Vector Labs) or biotinylated donkey antibodies (goat anti-mouse IgG secondary antibody HRP; Cat. # HAF007; Novus Biologicals, Littleton, CO, USA) in the same buffer for 1 h at room temperature. After washing three times for 10 min, the membranes were placed in 0.01 M Tris–HCl buffer (pH 7.2). The immunocytochemical reaction was conducted using the avidin–biotin ABC imaging system (Vectatain Elite ABC Kit, Vector Labs). To identify the reaction product, a red substrate was used (VIP Substrate Kit, Cat. # SK-4600; Vector Labs). After color development, the membranes were washed in distilled water and dried. For quantification, the obtained blots were scanned using a Bio-Rad GS 670 densitometer (Microelisa Stripplate Reader, Bio-Rad, Hercules, CA, USA).

The protein concentration in the samples was determined by cording according to the method of Bradford (1976) using bovine serum albumin (BSA) as a standard. We used BSA (Sigma-Aldrich, Merc, Millipore, Sigma-Aldrich, Supelco, USA). The method is based on shifting the absorption spectrum of the Coomassie blue G-250 dye towards 595 nm when it has been bound to a protein. To prepare the dye, 10 mg of Coomassie blue G-250 was dissolved with vigorous stirring in 5 mL of 95% ethanol and mixed with 10 mL of 85% phosphoric acid. The mixture was added to 100 mL H_2_O and filtered to remove insoluble dye. The protein concentration in the samples was determined in three analytical repetitions. To construct a calibration curve, solutions (5–100 μg/mL) were prepared from a standard BSA solution. A total of 1 mL of each BSA sample or calibration solution was mixed with 1 mL of Coomassie blue G-250 (Thermo Fisher Scientific, Waltham, MA, USA). Samples were stored in the dark at room temperature for 20 min. The optical density of the samples contained in a 1 mL glass cuvette was measured using a densitometer (Microelisa Stripplate Reader, Bio-Rad, Hercules, CA, USA) at a wavelength of 595 nm. The protein concentration (mg/mL) in the sample was calculated according to the calibration curve. Beta-actin with a molecular weight of 43 kDa was used as an additional loading control.

The molecular weight of GS was compared with prestained molecular weight markers (Sigma); this corresponded to the 37 kDa isoform of the protein. The molecular weight of the DC containing sample was compared to the prestained molecular weight markers (Sigma); this corresponded to 43 kDa. The molecular weight of PCNA was compared with prestained molecular weight markers (Sigma); this corresponded to 34 kDa. The molecular weight of Vim was compared with prestained molecular weight markers (Sigma); this corresponded to 57 kDa. The molecular weight of Nes was compared with prestained molecular weight markers (Sigma); this corresponded to 50 kDa. The concentration of GS, Vim, Nes, DC, and PCNA in the brain homogenates was determined.

#### 4.2.4. Immunohistochemistry

To study the proliferative activity and expression of glutamine synthetase, doublecortin, nestin, and vimentin in the cerebellum and brainstem of the rainbow trout, *O. mykiss*, immunoperoxidase labeling was performed on frozen, free-floating brain sections. Monoclonal mouse antibodies against GS (Catalog # ab64613, Abcam, Cambridge, UK; Catalog, diluted 1:300), PCNA (Catalog # Sc-53407, Santa Cruz Biotech, USA; diluted 1:300), doublecortin (Catalog # CO613 sc-271390, Santa Cruz Biotechnology, CA, USA, diluted 1:300), vimentin (Catalog # 3B4 ab28028, Abcam, Cambridge, UK; diluted 1:300), and nestin (Catalog # 2C1.3A11 ab18102, Abcam, Cambridge, UK; diluted 1:300) were used on 50 µm thick frontal sections, incubated in situ at 4 °C for 48 h. The immunocytochemical reaction was conducted using the avidin–biotin ABC imaging system (Vectatain Elite ABC Kit, Vector Labs). A red substrate (VIP Substrate Kit, Vector Laboratories, Burlingame, CA, USA) was used to identify the reaction products in accordance with the manufacturer’s recommendations. 

The brain sections were placed on glass slides coated with polylysine (BioVitrum, St. Petersburg, Russia) and left to dry completely. Cerebellar sections were additionally stained with 0.1% methyl green solution (Bioenno, Lifescience, Santa Ana, CA, USA, Cat. No. 003027) to identify immunonegative cells. Color development was monitored under a microscope. Sections were washed in three changes of distilled water for 10 s and differentiated for 1–2 min in 70% alcohol solution and then for 10 s in 96% ethanol and 10 s in 100% ethanol. The brain sections were dehydrated according to the standard procedure: they were placed in two xylene shifts of 15 min each and then placed in Bio-optica medium (Milano, Italy) under coverslips.

#### 4.2.5. Double Immunofluorescent Labeling 

To identify the expression patterns of PCNA and Vim, GS, and Nes, the following reagents were used: rabbit polyclonal antibodies against PCNA (Cat. No. PAA591Mi01, Cloud-Clone Corp. (CCC, Wuhan, China)) and mouse monoclonal antibodies against Vim (Catalog # 3B4 ab28028, Abcam, Cambridge, UK; diluted 1:50), GS (Catalog # ab64613, Abcam, Cambridge, UK; Catalog, diluted 1:50), and Nes (Catalog # 2C1.3A11 ab18102, Abcam, Cambridge, UK; diluted 1:50). In each series of experiments, a cocktail of mouse monoclonal antibodies against Vim, GS, or Nes in a 1:50 dilution was incubated together with polyclonal antibodies against PCNA in a 1:50 dilution used on 50 μm thick frontal sections, incubated in situ at 4 °C for 48 h. The resulting brain sections were pre-incubated in 0.1 M PBS, supplemented with 0.1% Tween-20 (Sigma-Aldrich, St. Louis, MO, USA, Cat. No. P9416) for 5 min. Then, they were incubated with 0.1 M PBS, with the addition of 0.3% Triton X-100 (Sigma-Aldrich, St. Louis, MO, USA, Cat. No. T8787) for 30 min at the room temperature. Afterwards, they were washed 3 times for 5 min with 0.1 M PBS, with the addition of 0.1% Tween-20. Then, the sections were blocked by immersing in 1% bovine serum albumin solution (BSA) (Sigma-Aldrich, St. Louis, MO, USA, Cat. No. B6917), supplemented with 0.1% Tween-20 for 2 h at room temperature. The brain sections were incubated with primary antibodies at 4 °C for 48 h, washed three times in 0.1 M PBS + 0.1% Tween-20 for 5 min, incubated with donkey anti-mouse conjugated secondary antibody Alexa 546 (Invitrogen, New York, NY, USA, dilution 1:200) and goat anti-rabbit conjugated secondary antibody Alexa 488 (Invitrogen, New York, NY, USA, dilution 1:200), washed with 0.1 M PBS and 0.1% Tween-20, and supplemented with 1% BSA and 0.1% Tween-20 for 3 min. Afterwards, the sections were embedded in the medium Fluoromount-G (Southern Biotech, Birmingham, AL, USA, Cat. No. 0100-20). Secondary antibodies Alexa 546 and Alexa 488 were also used alone, without primary antibodies, as the control. 

To identify the patterns of GS and Nes coexpression, the following reagents were used: rabbit polyclonal antibodies against GS (Cat. No. 5-35365) (Thermo Fisher Scientific, San Diego, CA, USA) and mouse monoclonal antibodies against Nes (Catalog # 2C1.3A11 ab18102, Abcam, Cambridge, UK; diluted 1:50). In each series of experiments, a cocktail of mouse monoclonal antibodies against Nes in a 1:50 dilution was incubated together with rabbit polyclonal antibodies against GS in a 1:50 dilution in 0.1% BSA, incubated at 4 degrees Celsius overnight. The resulting brain sections were pre-incubated in 0.1 M PBS, supplemented with 0.1% Tween-20 (Sigma-Aldrich, St. Louis, MO, USA, Cat. No. P9416) for 5 min. Then, they were incubated with 0.1 M PBS, with the addition of 0.1% Triton X-100 (Sigma-Aldrich, St. Louis, MO, USA, Cat. No. T8787) for 15 min at room temperature. Afterwards, they were washed 3 times for 5 min with 0.1 M PBS, with the addition of 0.1% Tween-20. Then, the sections were blocked by immersing in 1% bovine serum albumin solution (BSA) (Sigma-Aldrich, St. Louis, MO, USA, Cat. No. B6917), supplemented with 0.1% Tween-20 for 2 h at room temperature. The brain sections were incubated with a cocktail of primary antibodies at 4 °C for 48 h, washed three times in 0.1 M PBS + 0.1% Tween-20 for 5 min, incubated with a cocktail of donkey anti-mouse conjugated secondary antibody Alexa 546 (Invitrogen, New York, NY, USA, dilution 1:200) and anti-rabbit conjugated secondary antibody Alexa 488 (Invitrogen, New York, NY, USA, dilution 1:200), washed with 0.1 M PBS and 0.1% Tween-20, and supplemented with 1% BSA and 0.1% Tween-20 for 3 min. Afterwards, the sections were embedded in the medium Fluoromount-G (Southern Biotech, Birmingham, AL, USA, Cat. No. 0100-20). Secondary antibodies Alexa 546 and Alexa 488 were also used alone, without primary antibodies, as the control. 

#### 4.2.6. Microscopy

Visualization of the cells’ bodies with their morphological parameters and morphometric analysis (measurements of the greater and lesser diameters of the soma) were carried out under a Zeiss Axiovert 200M fluorescence motorized phase contrast microscope equipped with an ApoTome fluorescence module and AxioCam MRM and AxioCam HRC digital cameras (Carl Zeiss, Oberkohen, Germany). The material was analyzed using the AxioVision software http://ckp-rf.ru/ckp/3307/ (e.g., accessed on 1 April 2022). Measurements were performed at 100×, 200×, and 400× magnifications and in several randomly selected fields of view for each study area. The number of labeled cells per field of view was counted at a magnification of 200×. Micrographs of the sections were taken with an Axiovert 200 digital camera. The material was processed using the Axio Imager program and the Corel Photo-Paint 17 graphics editor.

#### 4.2.7. Densitometry

The optical density (OD) of immunohistochemical (IHC) labeling products in neuronal bodies and immunopositive granules was measured using the Axiovert 200-M microscope software http://ckp-rf.ru/ckp/3307/ (e.g., accessed on 1 April 2022). For this purpose, the Wizard program conducted a standard assessment of optical density for 5–7 sections, choosing 10–15 intensely/moderately labeled and immunonegative cells of the same type for analysis. Then, the average value of optical density for each type of cell was subtracted from the maximum value of optical density for immunonegative cells (background), and the actual values were expressed in relative units of optical density (UOD). Optical density (OD) in immunopositive cells was categorized on the following scale: high (180–130 units of optical density (UOD), designated as +++), moderate (130–80 UOD, ++), weak (80–40 UOD, +), and low (less than 40 UOD, corresponding to negative cells); the initial OD value was measured on the control mounts. The average OD value for each type of cell was subtracted from the maximum OD value for immunonegative cells (background) to obtain the actual value in relative optical density units (ODU). Based on the densitometric analysis data, various levels of PCNA, DC, GS, Vim, and Nes activity in cells were determined. These data, along with the dimensional characteristics, were used for cell typing on the basis of the previously developed classification of cells in the pallial zones of the *O. mykiss* telencephalon [3] which are formed during the period of constitutive and reparative neurogenesis.

#### 4.2.8. Statistical Analysis

The quantitative processing of morphometric data of IHC labeling was performed using the Statistica 12, Microsoft Excel 2010, and STATA software packages (StataCorp. 2012, Stata Statistical Software: Release 12. College Station, TX: StataCorp LP, USA). All data are presented as mean ± standard deviation (M ± SD) and were analyzed using the SPSS software application (version 16.0; SPSS Inc., Chicago, IL, USA). All changes in the group were compared using the Student–Newman–Keuls test or a one-way analysis/two-way of analysis variance (ANOVA, Chicago, IL, USA) with Bonferroni’s correction. The normality of the distribution was tested using a normal probability graph. Differences of values at *p* ≤ 0.01 and *p* ≤ 0.05 were considered statistically significant.

## 5. Conclusions

Concerning the investigation objective, it should be noted that, since the molecular profile of aNSPCs in rainbow trout and other fishes is characterized by high heterogeneity, it was interesting to study the distribution of some structural proteins such as vimentin (Vim), nestin (Nes), and the enzyme glutamine synthetase (GS), which are found in NSPCs of both embryonic type (neuroepithelial cells) and adult type (radial glia). It is well known that the cells containing these molecular markers and having the ability to proliferate in fish are considered as NSCs [12,18,35,38]. The IHC labeling with PCNA revealed the areas in the rainbow trout cerebellum and brainstem containing proliferating cells, which coincide with the areas expressing Vim, Nes, and GS. Double immunolabeling confirmed the colocalization of these markers in the brainstem in the PVZ (Vim/PCNA and GS/PCNA) and SMZ (Nes/PCNA) cell populations. The Vim, Nes, and GS expressions in trout have a number of specific features and differ from the expressions of these markers in mammals and other fish species. In particular, the distribution of GS in trout is localized mainly in the cells’ bodies and RG processes, while in the mammalian brain, GS mainly labels cellular bodies [1,2,10]. Vim, in contrast, was almost not found in RG processes in the trout cerebellum and brainstem, while it was localized in bodies of large neurons, as well as in undifferentiated NE-type cells in the periventricular zone (PVZ) and submarginal zone (SMZ) of proliferation. Compared to Vim, Nes constitutes the predominant structural protein in the trout brain. The identification of cells expressing DC, Vim, and Nes in the cranial nerve IX–X nuclei of the trout is consistent with data on mammals [46]. Thus, the neurons expressing DC, Vim, and Nes were found in the trout brainstem in the nuclei of pairs IX–X of the parasympathetic column of cells, which corresponds to the data on the vagal complex of the brain of adult rodents, in which a neurogenic niche was found in this area. 

The most important feature of the localization of proliferating cells in the trout brainstem and cerebellum is the presence of such cells not only in the primary zone of proliferation (PVZ) but also in the SMZ of the brainstem, which distinguishes trout from other studied fish species. The results of the IHC study of DC distribution in the trout cerebellum and brainstem have shown a high level of expression of this marker in various cell populations. This may indicate (i) a high production of adult neurons in the cerebellum and brainstem of adult trout, which differs from data on zebrafish, and (ii) a high plasticity of neurons in the cerebellum and brainstem of trout. We suggest that the source of new cells in the trout brain, along with the PVZ and SMZ containing proliferating cells, may constitute local neurogenic niches also containing the silent (PCNA-negative) cells which express aNSC markers. 

Studying the properties of aNSPCs using the salmon model provides new data on the organization of neurogenic zones in various parts of the brain containing adult-type stem cells, with an emphasis on their development, origin, cell lines, and proliferation dynamics [4,5]. Currently, the molecular signatures of these populations during homeostasis repair in the brains of vertebrates are only beginning to be studied [2,6,7,8]. Beyond the brain, the regenerative plasticity of adult stem cells/progenitor cells and their biological significance remain poorly understood. This is especially true for the caudal parts of the brain, in particular, the brainstem and cerebellum, where neural stem cells (NSCs) have been found in *Danio rerio* [2,6], as well as in juvenile chum salmon [4] and masu salmon [5]. The study of the properties of these cells is important from the point of view of the potential of their possible use in clinical practice in the treatment of neurodegenerative diseases, in particular, Alzheimer’s disease, Parkinson’s disease, traumatic brain injuries, and stroke. Effective post-traumatic repair in the cerebellum of juvenile chum salmon [13,14] and an increase in the number of GS+ and Aromatase-expressing, hydrogen sulfide-producing aNSPCs after long-term injury, as well as repeated injury, indicate effective neuroprotective mechanisms contributing to the successful recovery of the brain of salmon fish after trauma.

## Figures and Tables

**Figure 1 ijms-25-05595-f001:**
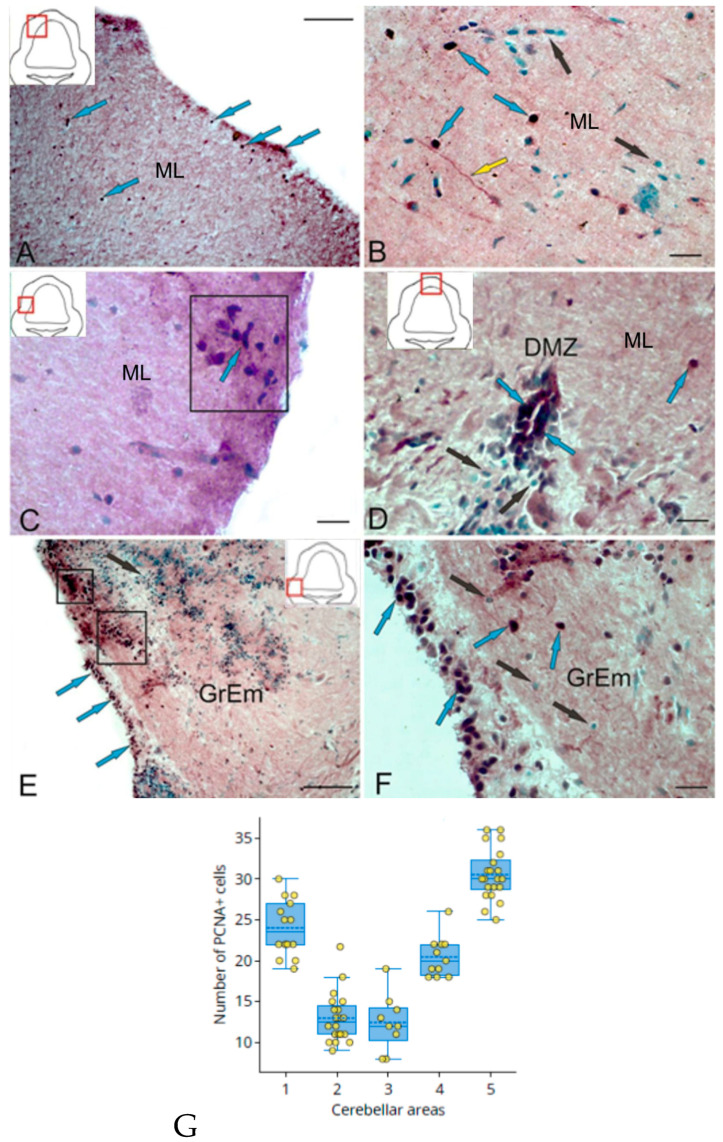
Proliferating cell nuclear antigen (PCNA) in the cerebellum of a rainbow trout, *Oncorhynchus mykiss*. (**A**)—General view of PCNA distribution in the dorso-lateral part of the corpus cerebellum; ML—molecular layer; blue arrows indicate immunopositive cells. (**B**)—Dorso-lateral zone at higher magnification; black arrows indicate PCNA-immunonegative cells; blue arrows indicate PCNA+ cells; yellow arrow indicates PCNA+ radial glia fibers. (**C**)—Clusters of PCNA+ cells in the superficial layers of the ML (in black rectangle); other designations see above. (**D**)—Dorsal matrix zone (DMZ) containing PCNA+ (blue arrows) and -negative (black arrows) cells. (**E**)—Population of PCNA+ cells (in black rectangle) in granular eminence (GrEm). (**F**)—At a higher magnification. Scale bars: (**A**,**E**) 100 µm; (**B**–**D**,**F**) 20 µm. (**G**)—Comparative distribution of PCNA+ labeled cells in different areas of the cerebellum; 1—superficial part of molecular layer; 2—parenchymal part of molecular layer; 3—cluster of cells; 4—DMZ; 5—granular eminence; (mean ± SD), where M is the mean and SD is the standard deviation (*n* = 15–20 in each group). Data analyzed using unpaired *t*-test (two-tailed).

**Figure 2 ijms-25-05595-f002:**
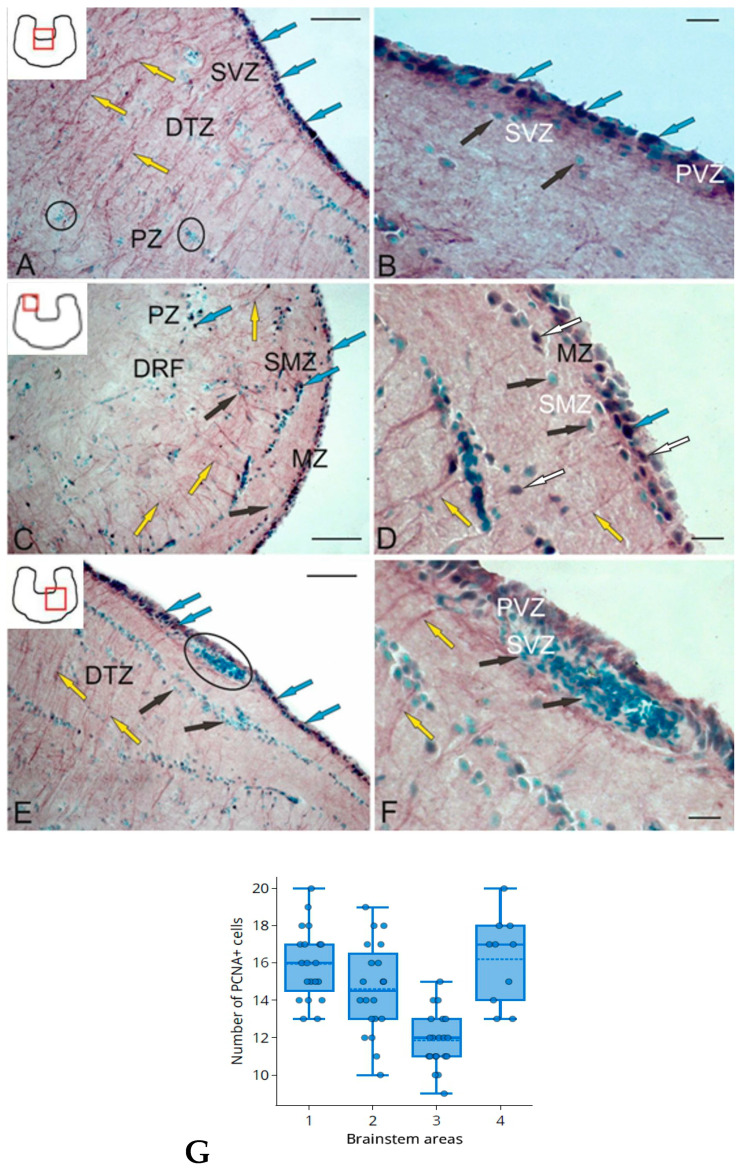
PCNA in the brainstem of a rainbow trout, *Oncorhynchus mykiss*. (**A**)—Distribution of PCNA in the brainstem at the isthmus level (Is); blue arrows indicate PCNA+ cells in the PVZ; PCNA+ fibers of RG are shown by yellow arrows; SVZ—subventricular zone; PZ—parenchymal zone; aggregations of PCNA-negative cells are outlined by black ovals. (**B**)—A fragment of the periventricular region at a higher magnification; black arrows indicate PCNA-negative cells. (**C**)—PCNA+ cells at the dorsal reticular formation (DRF) level; MZ—marginal zone; SMZ—submarginal zone; PZ—parenchymal zone. (**D**)—A fragment of the marginal zone at a higher magnification; white arrows indicate moderately labeled PCNA+ cells in SMZ and MZ. (**E**)—Aggregations of PCNA-negative cells in the SVZ of the dorsal tegmental zone (DTZ), outlined by an oval. (**F**)—At a higher magnification. Scale bars: (**A**,**C**,**E**) 100 µm; (**B**,**D**,**F**) 20 µm; (**G**)—Comparative distribution of the number of PCNA+ cells in different areas of the brainstem; 1—PVZ; 2—superficial layer of brainstem; 3—cluster of cells in the superficial layer; 4—parenchyma of brainstem; (mean ± SD), where M is the mean and SD is the standard deviation (*n* = 15–20 in each group). Data analyzed using unpaired *t*-test (two-tailed).

**Figure 3 ijms-25-05595-f003:**
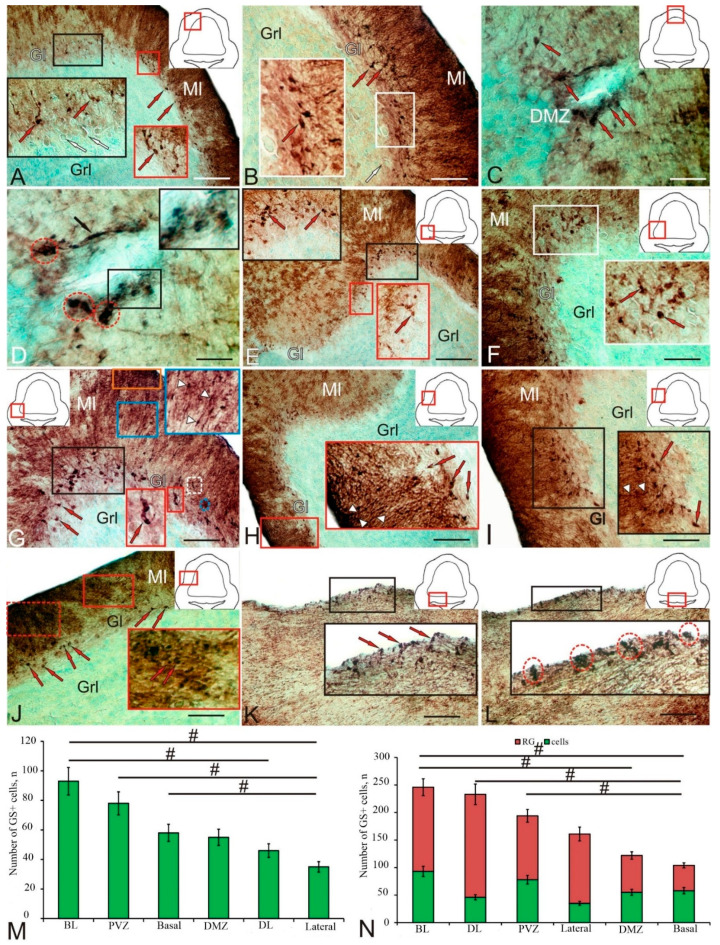
Glutamine synthetase (GS) in the cerebellum of a rainbow trout, *Oncorhynchus mykiss*. (**A**)—General view of GS immunolocalization patterns in the dorso-lateral part of the trout cerebellum: Ml—molecular layer; Gl—ganglion layer; Grl—granular layer; the pictogram shows the zones of the dorso-lateral cerebellum, aggregations of GS+ cells, intensely labeled cells (red arrows in the inset outlined by red rectangle), and diffuse aggregations of immunopositive and -negative cells (white arrows)—inset outlined by black rectangle. (**B**)—Dorso-lateral region (Dl) at higher magnification; inset (outlined by white rectangle) shows projection of eurydendroid cells (EDCs) (white arrows). (**C**)—Dorsal matrix zone (DMZ); the pictogram shows the DMZ and intensely labeled GS cells of the neuroepithelial (NE) type (red arrow). (**D**)—An enlarged DMZ fragment with elongated NE cells (black arrow), aggregation of GS+ cells with small processes in the middle part of the DMZ (black inset), and clusters of undifferentiated GS+ cells (in red dotted ovals) in the basal part of the DMZ. (**E**)—General view of distribution of GS in the rostro-basal part (red square in the pictogram) of the cerebellum, an aggregation of basal intensely labeled GS+ cells (black inset), and a group of moderately labeled GS+ cells (red inset). (**F**)—At higher magnification; the baso-lateral region (in the pictogram); intensely labeled GS+ cells of different types (inset in the white rectangle). (**G**)—The caudal baso-lateral zone of the cerebellum (pictogram); an aggregation of GS+ astrocyte-like cells in the ganglionic layer (GL, in black rectangle); areas of GS+ radial glia (white arrowheads) in the molecular layer (in blue inset); dense accumulation of GS+ undifferentiated cells in the superficial part of the ML (in orange box); a cluster of two GS+ astrocyte-like cells (red inset); a dense cluster of undifferentiated GS+ cells (dashed blue oval); a diffuse cluster of GS+ undifferentiated cells (white dotted square). (**H**)—The rostral lateral area of the CC (pictogram); an area with intensely labeled GS+ NE cells (white arrowheads in the inset). (**I**)—The caudo-lateral region of the CC (inset). (**J**)—Aggregations of GS+ astrocyte-like cells in the ML (red inset); an aggregation of GS+ radial glia is shown in the red dotted rectangle. (**K**)—Diffuse aggregations of GS+ undifferentiated cells in the dorso-lateral zone of the IV ventricle roof (inset). (**L**)—Clusters of undifferentiated GS+ cells (outlined by red dotted ovals) in the dorso-medial zone of the IV ventricle roof (inset). Scale bars: (**A**,**E**,**H**,**I**) 200 µm; (**B**,**F**,**G**,**J**–**L**) 100 µm; (**C**,**D**) 50 µm. (**M**)—Ratio of GS+ labeled cells in different areas of the cerebellum; significant intergroup differences # (*p* < 0.05) between groups of labeled cells in different areas of the cerebellum (*n* = 5 in each group), one-way ANOVA. (**N**)—Ratio of GS+ labeled astrocyte-like cells and radial glia in different areas of the cerebellum; significant intergroup differences # (*p* < 0.05) between groups of labeled cells in different areas of the cerebellum (*n* = 5 in each group), one-way ANOVA; number of immunopositive RG and cells in different areas of the cerebellum (mean ± SD), where M is the mean and SD is the standard deviation (*n* = 5 in each group).

**Figure 4 ijms-25-05595-f004:**
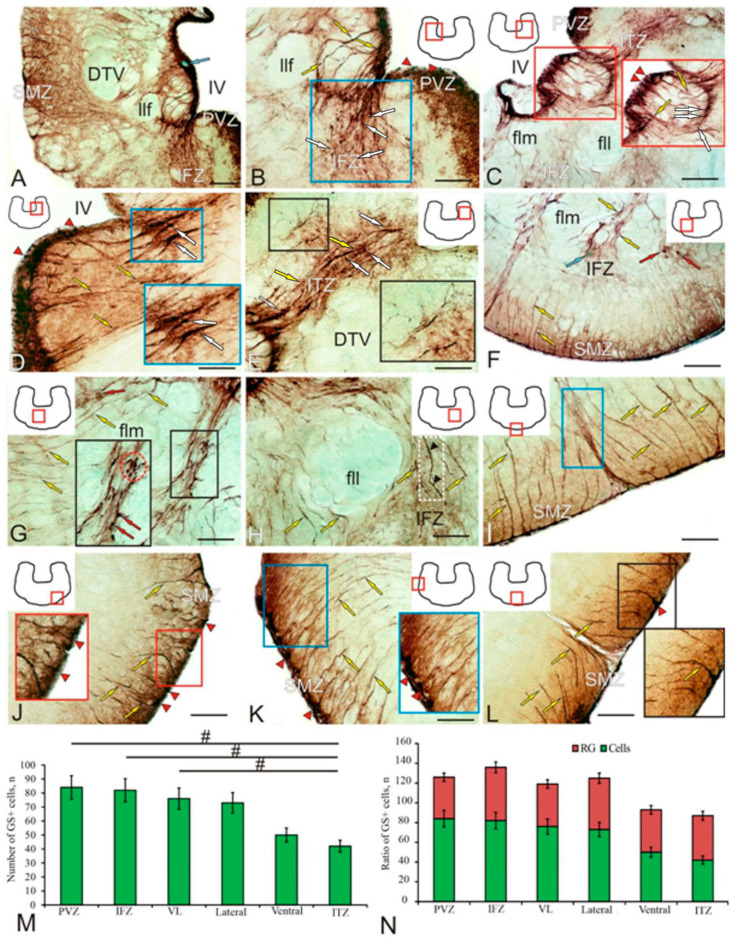
Glutamine synthetase (GS) in the brainstem of a rainbow trout, *Oncorhynchus mykiss*. (**A**)—General view of GS immunolabeling in the brainstem; PVZ—periventricular zone; SMZ—submarginal zone; IFZ—interfascicular zone; DTV—descending trigeminal tract; fll—lateral longitudinal fascicle; IV—fourth ventricle; the blue arrow indicates a vessel. (**B**)—IFZ (in the blue rectangle); white arrows indicate GS+ astrocyte-like cells; yellow arrows indicate GS+ fibers of radial glia; red arrowheads indicate neuroepithelial (NE) GS+ cells. (**C**)—Lateral wall of the IV ventricle (red inset); ITZ—intertrigeminal zone; flm—medial longitudinal fascicle. (**D**)—An enlarged fragment of the lateral wall (blue inset), areas with intensely labeled astrocyte-like cells (white arrows); (**E**)—The central part of the intertrigeminal zone with GS+ fibers (inset); immunopositive astrocyte-like cells (white arrows); GS+ RG fibers (yellow arrows). (**F**)—RG fibers in the SMZ (yellow arrows); a vessel (blue arrow); GS+ astrocyte-like cells (red arrows). (**G**)—Section through the IFZ (inset) containing a cluster of GS+ astrocyte-like cells (in the red dotted oval) and single GS+ astrocyte-like cells (red arrows). (**H**)—GS+ RG fibers around fll; a fragment of an immunopositive RG (black arrowheads in white dotted rectangle). (**I**)—GS+ RG in the region of the central raphe (in blue rectangle). (**J**)—The ventro-lateral region of the SMZ; GS+ cells in the superficial layers of the SMZ (inset); GS+ RG (yellow arrows). (**K**)—The lateral region of the brainstem; immunopositive RG (inset). (**L**)—The paramedian region of the brainstem; thick RG fibers (inset). Scale bars: (**A**) 200 µm; (**B**–**L**) 100 µm. (**M**)—Comparative distribution of GS+ cells in different areas of the brainstem; intergroup differences between the ITZ and PVZ, IFZ, and ventro-lateral part # (*p* < 0.05), one-way ANOVA (*n* = 5 in each group). (**N**)—Comparative distribution of the total number of NE and RG GS+ cells in different areas of the brainstem (mean ± SD), where M is the mean and SD is the standard deviation (*n* = 5 in each group).

**Figure 5 ijms-25-05595-f005:**
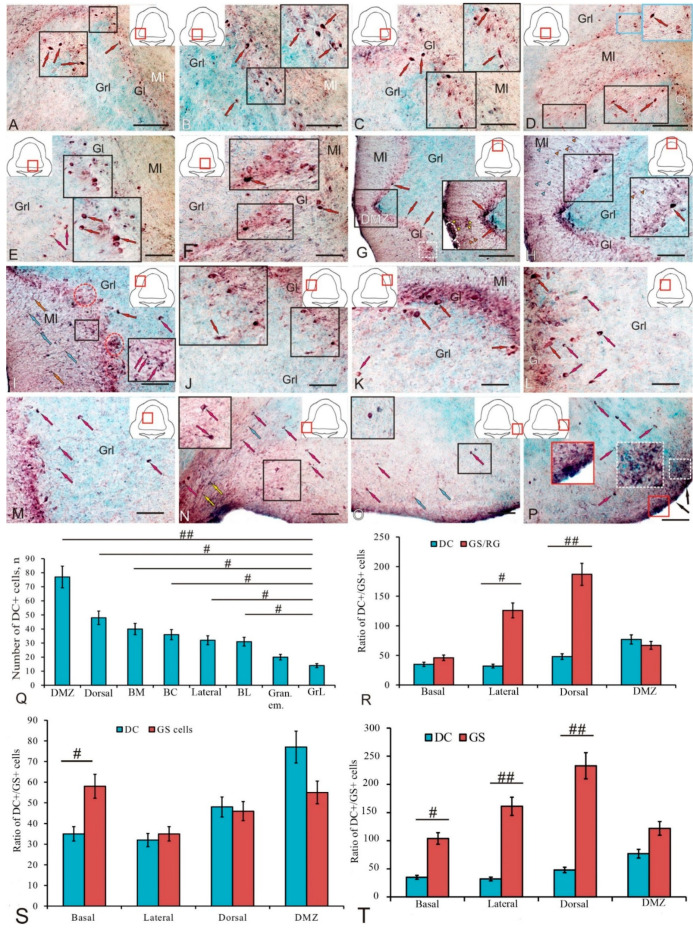
Immunolocalization of doublecortin (DC) in different regions of the cerebellum of a rainbow trout, *Oncorhynchus mykiss*. (**A**)—General view of DC immunolocalization patterns in the baso-lateral zone of the trout cerebellum (pictogram); immunopositive cells of the ganglion layer (Gl) (inset, red arrows). (**B**)—Heterogeneous DC immunolabeling in projection neurons (red arrows) of the ganglion layer (inset). (**C**)—Heterogeneous immunolabeling of cells in the baso-lateral region of the cerebellum (inset); ectopic EDCs (red arrow); (**D**)—A paramedian region of the basal part containing DC+ cells (black inset) in the dorsal part of the Ml invagination with intensely labeled EDCs in the basal part of the invagination (blue inset). (**E**)—DC immunolocalization in the central zone (inset); Gl cells and adjacent areas containing heterogeneous immunolabeling (red arrows). (**F**)—An enlarged fragment of the paramedian region (inset) containing heterogeneous DC immunolabeling in cells (red arrows). (**G**)—DC immunolocalization in the DMZ (inset); immunopositive NE cells in the basal part of the DMZ (red arrows); RG fibers (yellow arrowheads) and dense clusters of superficially located undifferentiated immunopositive cells (white dotted oval); (**H**)—DC+ cell (red arrow) in the dorsal region of the cerebellum adjacent to the DMZ; thin RG fibers (blue arrowheads) and thick fibers (orange arrowheads). (**I**)—The dorso-lateral region of the cerebellum with an aggregation of small DC+ neuroblasts (inset) in the basal part of the Gl, clusters of heterogeneously labeled DC (outlined by red dotted ovals), RG fibers in the Ml (blue arrows), and weakly immunolabeled DC cells in the Ml (orange arrows). (**J**)—Large projection EDCs in the lateral region of the cerebellum (inset); an ectopic cell of atypical localization is indicated by a red arrow. (**K**)—DC-immunopositive pear-shaped Purkinje neurons and DC+ cells in the granular layer (Grl) (red arrows) in the lateral region of the cerebellum. (**L**)—Large DC+ granular cells with processes and smaller cells of the periganglionic region in the Grl (pictogram). (**M**)—General view of DC immunolocalization in the central granular region (pictogram). (**N**)—DC+ neurons in granular eminence (inset) and numerous DC+ neurons (pink arrows) and RG (yellow arrows) in the region of the cerebellar peduncles. (**O**)—DC+ cells of granular type in granular eminence (inset); other designations as in (**N**). (**P**)—Superficial clusters of DC+ cells (red inset) and in the subsurface layer (white dotted inset) in granular eminence. Scale bars: (**A**,**G**) 200 µm; (**B**–**F**,**H**–**P**) 100 µm. (**Q**)—Comparative distribution of DC+ cells in different areas of the trout cerebellum (mean ± SD); significant intergroup differences between Grl and dorsal, baso-medial, baso-central, lateral, and baso-lateral regions # (*p* < 0.05); between Grl and DMZ ## (*p* < 0.01) (*n* = 5 in each group); one-way ANOVA. (**R**)—Comparative distribution of DC+ and GS+ RG in different parts of the trout cerebellum; significant intergroup differences in the number of cells in the lateral part # (*p* < 0.05) and in the dorsal part ## (*p* < 0.01) (*n* = 5 in each group), one-way ANOVA. (**S**)—Comparative distribution of DC+ and GS+ NE cells in different areas of the trout cerebellum (mean ± SD), where SD is the standard deviation (*n* = 5 in each group). (**T**)—Comparative distribution of DC+ and the total number of GS+ cells and RG in the trout cerebellum; significant intergroup differences in the number of cells in the basal part # (*p* < 0.05), in the lateral and dorsal parts ## (*p* < 0.01) (*n* = 5 in each group), one-way ANOVA.

**Figure 6 ijms-25-05595-f006:**
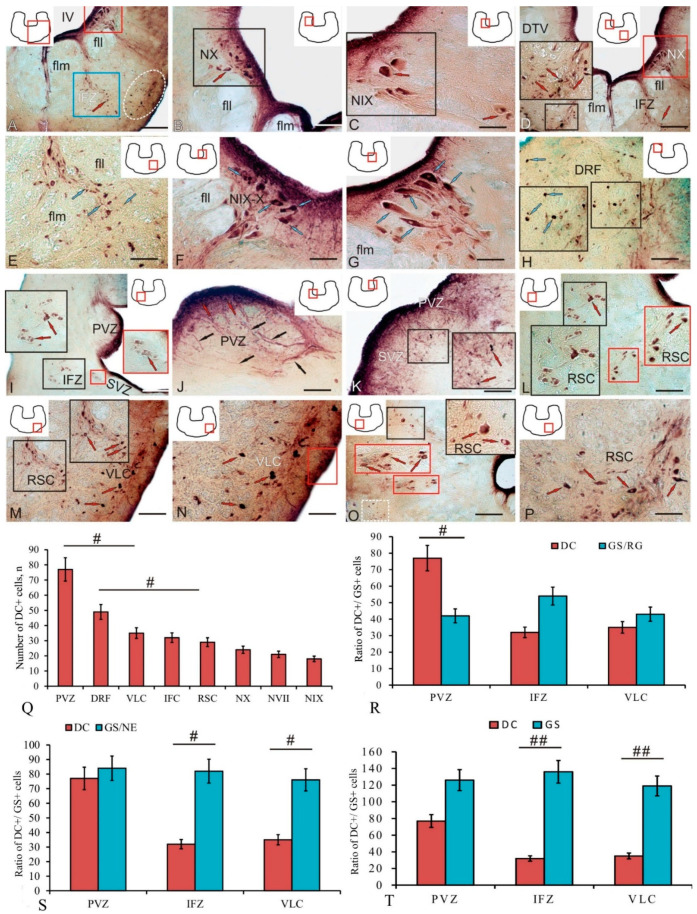
Doublecortin (DC) immunolabeling in the brainstem of a rainbow trout, *Oncorhynchus mykiss*. (**A**)—General view of DC distribution in neurons of the parasympathetic cell column (in red rectangle), in the reticular formation (in the blue square), and cells of the ventro-lateral region (in white dotted oval). (**B**)—DC+ motoneurons (red arrow) in the nucleus of the X nerve (in black rectangle). (**C**)—DC+ neurons in the nucleus of the glossopharyngeal nerve (in black rectangle). (**D**)—General view of DC immunolabeling in cells of the interfascicular zone (IFZ) (black inset) and nucleus of the X nerve (in red rectangle). (**E**)—DC+ neurons (blue arrows) in the IFZ at a higher magnification. (**F**)—DC+ neurons in the nucleus tractus solitaris (red arrow) and dorsal parasympathetic column (blue arrows). (**G**)—DC+ neurons in the nucleus of the facial nerve. (**H**)—Heterogeneous population (inset) of DC+ cells (blue arrows) in the dorsal reticular formation (DRF). (**I**)—Distribution of DC in the caudal part of the brainstem, IFZ (black inset), and subventricular zone (SVZ) (red inset). (**J**)—Ventro-lateral part of the floor of the IV ventricle at higher magnification (pictogram); RG fibers (black arrows) of DC+ cells of the periventricular zone (PVZ) (red arrows). (**K**)—A lateral part of the wall of the IV ventricle (pictogram); DC+ cells of the parenchymal part of the SVZ (inset). (**L**)—Cells of the ventro-lateral part of the reticular formation; dorsal DC+ population of reticulospinal neurons (black inset) and lateral DC+ population of reticulospinal cells (RSCs) (red inset). (**M**)—Mutual arrangement of reticular DC+ cells (in black rectangles) and ventro-lateral DC+ cell population. (**N**)—ventro-lateral DC-immunopositive cells (red arrows) and submarginal cluster of DC+ cells (in red rectangles). (**O**)—Median DC+ cells (red inset) and ventro-lateral (black inset) immunopositive cells of the reticular formation. (**P**)—DC+ RSCs (red arrows) at a higher magnification. Scale bars: (**A**,**B**,**D**,**I**,**O**) 200 µm; (**C**,**E**–**H**,**K**–**P**) 100 µm; (**J**) 50 µm. (**Q**)—Comparative distribution of DC+ cells in different areas of the trout brainstem (mean ± SD), significant intergroup differences between the PVZ and the population of ventro-lateral cells (VLCs) and the DRF and interfascicular cells (IFCs) # (*p* < 0.05) (*n* = 5 in each group), one-way ANOVA. (**R**)—Comparative distribution of DC+ and GS+ RG cells in different areas of the trout brainstem (mean ± SD), where SD is the standard deviation (*n* = 5 in each group). (**S**)—Comparative distribution of DC+ and total number of GS+ NE cells in different areas of the trout brainstem; significant intergroup differences in the number of cells in the IFZ and VLC population # (*p* < 0.05) (*n* = 5 in each group), one-way ANOVA. (**T**)—Comparative distribution of DC+ and total number of GS+ cells and RG in the trout brainstem, significant intergroup differences in the number of cells in the IFZ and VLC population ## (*p* < 0.01) (*n* = 5 in each group), one-way ANOVA.

**Figure 7 ijms-25-05595-f007:**
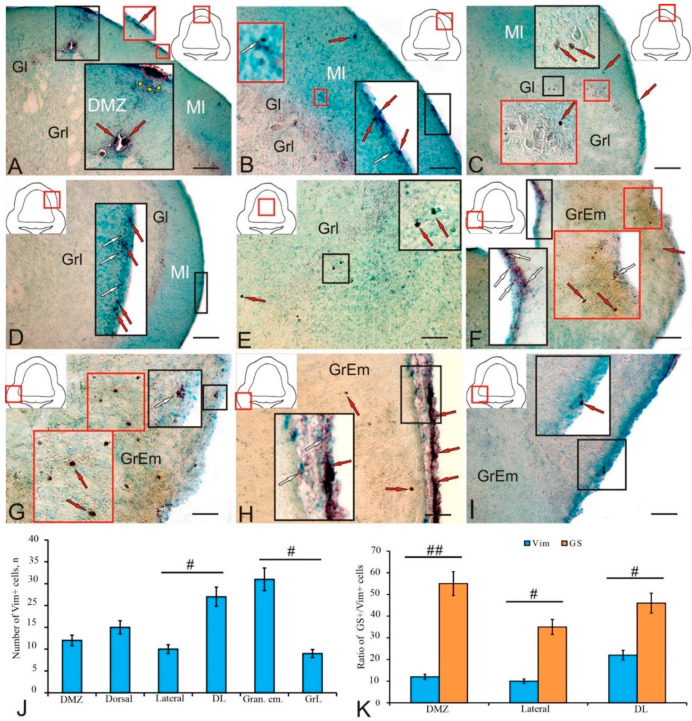
Vimentin (Vim) in the cerebellum of a rainbow trout, *Oncorhynchus mykiss*. (**A**)—Vim immunolabeling in the dorsal matrix zone (DMZ) (black inset); neuroepithelial (NE)-type cells (red arrows); a dense cluster of superficially located undifferentiated Vim+ cells (in red dotted oval); diffusely located cells (yellow arrowheads); a cluster of superficial undifferentiated cells in the dorsal region (red inset). (**B**)—Vim+ cells in the dorso-lateral part of the corpus cerebellum (CC) (black inset); immunopositive granules (white arrow); diffuse accumulation of small, intensely labeled NE cells (red inset, indicated by white arrow). (**C**)—Vim+ pear-shaped cells in the caudal region of the ganglionic layer (Gl) of the cerebellum (black inset, arrows), as part of the periganglionic zone (red inset). (**D**)—Immunopositive NE-type cells in the superficial (red arrows) and subsurface (white arrows) layers of the lateral part of the cerebellum (inset). (**E**)—Vim+ cells in the central part of the granular layer (inset). (**F**)—Vim+ cells in outer layers of granular eminence (GrEm) (black inset, white arrows) and the parenchymal zone of granular eminence (red inset, red arrows). (**G**)—Cluster of immunopositive NE cells in the outer zone of GrEm (black inset, white arrow) and Vim+ granular cells in the parenchyma of GrEm (red inset, red arrows). (**H**)—Dense extensive clusters of immunopositive cells (inset, red arrows) and granules (white arrows) in the surface layer of GrEm; (**I**)—Single immunopositive cells in the surface layer (inset, red arrow) of GrEm. Scale bars: (**A**,**C**,**D**,**F**) 200 µm; (**B**,**E**,**G**–**I**) 100 µm. (**J**)—Comparative distribution of Vim+ cells in different parts of the trout cerebellum; significant intergroup differences in the number of cells in the lateral and dorso-lateral parts of the CC and in GrEm and Grl # (*p* < 0.05) (*n* = 5 in each group), one-way ANOVA. (**K**)—Comparative distribution of Vim+ cells and the total number of GS+ cells and RG in different parts of the trout cerebellum; significant intergroup differences in the number of cells in the DMZ ## (*p* < 0.01), lateral, and dorso-lateral parts # (*p* < 0.05) (*n* = 5 in each group), one-way ANOVA.

**Figure 8 ijms-25-05595-f008:**
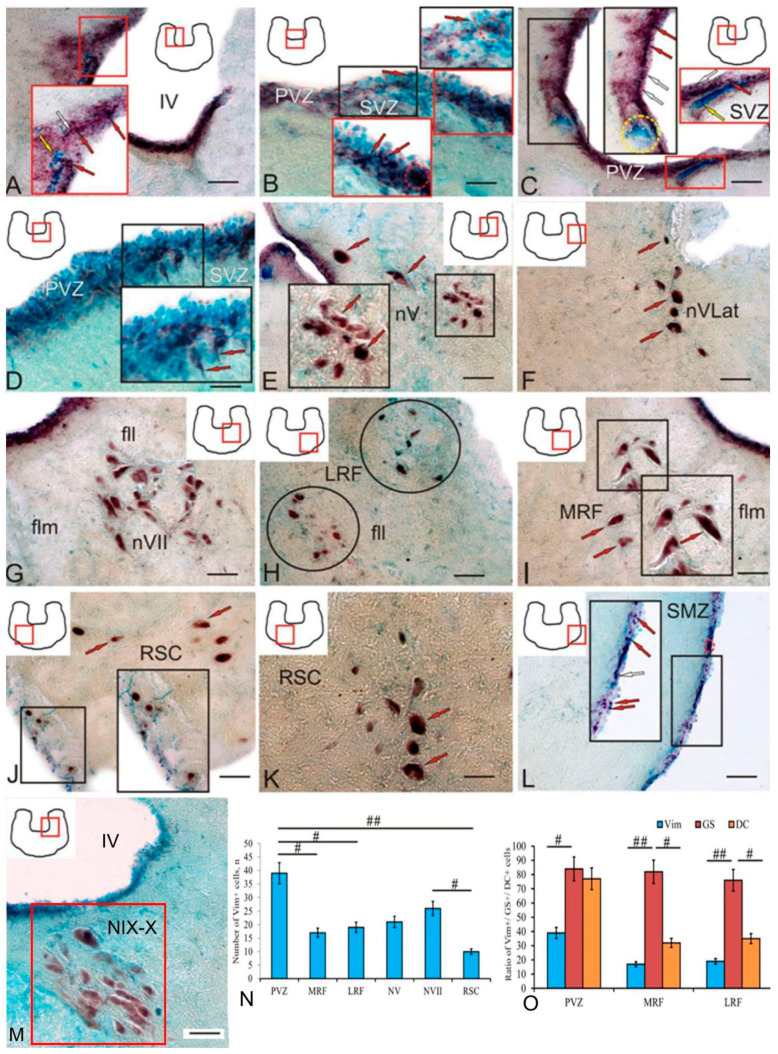
Vimentin (Vim) in the brainstem of a rainbow trout, *Oncorhynchus mykiss*. (**A**)—General view of Vim distribution in the periventricular zone (PVZ) of the lateral wall of the IV ventricle; aggregation of periventricular Vim+ NE-type cells (inset, red arrows); single immunopositive cells (white arrows); clusters of immunonegative cells (yellow arrows). (**B**)—Vim+ undifferentiated cells (red arrows in red inset) and their clusters (in red dotted oval) at the bottom of the IV ventricle; weakly labeled Vim+ NE cells (black inset, red arrow) in the PVZ; SVZ—subventricular zone. (**C**)—Dense clusters (red arrows) and sparser clusters (white arrows); clusters of immunonegative cells (in yellow dotted oval) in the lateral wall of the IV ventricle (black inset) and on the ventral wall (red inset) of the IV ventricle; elongated aggregation of immunonegative cells (yellow arrow); single Vim+ cells (red arrow); surface immunopositive cells (white arrow). (**D**)—Immunopositive cells with processes (inset, red arrows) in the SVZ of the ventro-lateral part of the IV ventricle. (**E**)—Single Vim+ neurons (red arrows) in the main nucleus of the V nerve (inset). (**F**)—Vim+ neurons (red arrows) in the lateral nucleus of the V nerve. (**G**)—Vim+ neurons (red arrows) in the nucleus of the VII nerve; fll—lateral longitudinal fascicle; flm—medial longitudinal fascicle. (**H**)—Populations of Vim+ neurons of the lateral reticular formation (LRF, in black ovals). (**I**)—Immunopositive interneurons (red arrows) and aggregations of large Vim+ neurons (inset) in the medial reticular formation (MRF). (**J**)—Populations of Vim+ reticulospinal neurons (red arrows) and ventro-lateral Vim+ cells (inset). (**K**)—Vim+ reticulospinal cells (red arrows) at a higher magnification. (**L**)—Clusters of immunopositive undifferentiated cells (inset, white arrow) and single cells (red arrow) in the submarginal zone (SMZ) of the trout brainstem. (**M**)—Vim+ motor neuron (red arrow) in the parasympathetic column and interneurons in the sensory part (white arrow) in the nucleus of IX-X nerves (in the red rectangle). Scale bars: (**A**,**E**–**L**) 200 µm; (**B**–**D**) 100 µm. (**N**)—Comparative distribution of Vim+ cells in different areas of the trout brainstem; significant intergroup differences in the number of cells in the PVZ and RSC population ## (*p* < 0.01), in the PVZ and MRF, in the PVZ and LRF # (*p* < 0.05), and in the NVII and RSC population # (*p* < 0.05) (*n* = 5 in each group), one-way ANOVA. (**O**)—Comparative distribution of Vim+, DC+, and GS+ cells in different areas of the trout brainstem; significant intergroup differences in the number of cells in the PVZ for Vim+ and GS+ cells # (*p* < 0.05) and in the MRF and LRF for Vim+ and GS+ cells ## (*p* < 0.01) and for GS+ and DC+ cells # (*p* < 0.05) (*n* = 5 in each group), two-way ANOVA.

**Figure 9 ijms-25-05595-f009:**
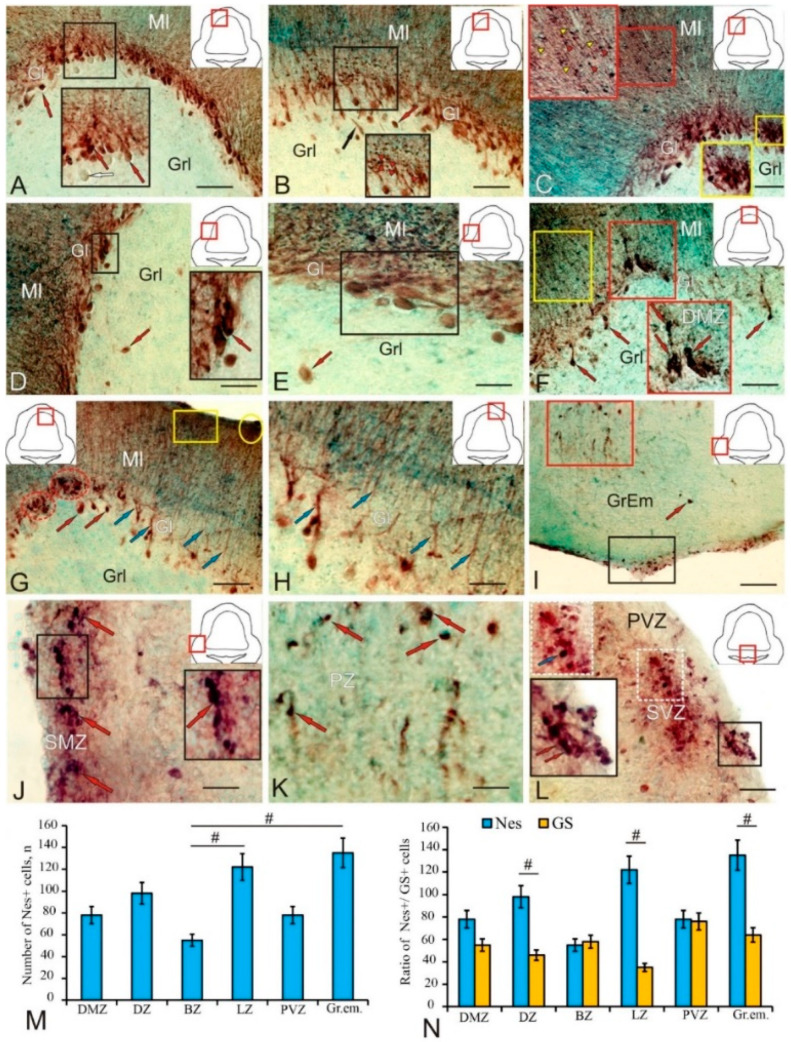
Nestin (Nes) in the cerebellum of a rainbow trout, *Oncorhynchus mykiss*. (**A**)—General view of Nes immunolabeling in the dorsal part of the trout cerebellum (inset); Nes+ Purkinje cells (red arrows) and Nes− cells (white arrows); Gl—ganglion layer; Ml—molecular layer; Grl—granular layer. (**B**)—Intensively labeled Nes+ granules (inset, red arrowheads) and immunopositive proximal parts of Purkinje cell dendrites (black arrow). (**C**)—Intensely labeled Nes+ cells in the Ml (red inset, red arrowheads) and radial fibers (yellow arrowheads); clusters of Nes+ cells in the Ml (yellow inset) of the dorso-lateral part of the corpus cerebellum (CC). (**D**)—Clusters of immunopositive cells in the Gl (inset) in the lateral part of the CC. (**E**)—Intensely labeled immunopositive cells (in the black rectangle) in the baso-lateral part of the Gl; Nes+ granule cell (red arrow). (**F**)—Immunopositive cells of the NE type in the DMZ (red inset); a fragment with Nes+ structures in the Ml (in yellow rectangle). (**G**)—Clusters of immunopositive cells (in red dotted ovals) in the Gl; bodies of Nes+ Purkinje cells (red arrows); proximal parts of immunopositive dendrites (blue arrows); surface populations of small undifferentiated Nes+ cells in the Ml (in yellow rectangle and oval). (**H**)—Enlarged fragments with immunopositive proximal dendrites (blue arrows). (**I**)—Surface population of intensely labeled Nes+ cells of the NE type (in black box) and an aggregation of parenchymal immunopositive granule cells (in the red box) in granular eminence (GrEm) (inset). (**J**)—Nes+ cells (red arrows) in the submarginal zone (SMZ, black inset) of GrEm; (**K**)—Nes+ granular cells (red arrows) in the parenchyma of GrEm. (**L**)—Clusters of superficial Nes+ cells (red arrows) in the periventricular zone (PVZ, black inset) and subventricular zone (SVZ, white dotted inset); blue arrow indicates an intensely labeled cell. Scale bars: (**A**–**D**,**F**,**G**,**I**) 200 µm; (**E**,**H**,**J**–**L**) 100 µm. (**M**)—Comparative distribution of Nes+ cells in different areas of the cerebellum; significant intergroup differences in the number of cells in the basal zone (BZ) and lateral zone (LZ) and in the BZ and GrEm # (*p* < 0.05) (*n* = 5 in each group), one-way ANOVA. (**N**)—Comparative distribution of Nes+ and GS+ cells in different areas of the trout cerebellum; significant intergroup differences in the number of Nes+ and GS+ cells in the dorsal zone (DZ), LZ, and GrEm # (*p* < 0.05) (*n* = 5 in each group), one-way ANOVA.

**Figure 10 ijms-25-05595-f010:**
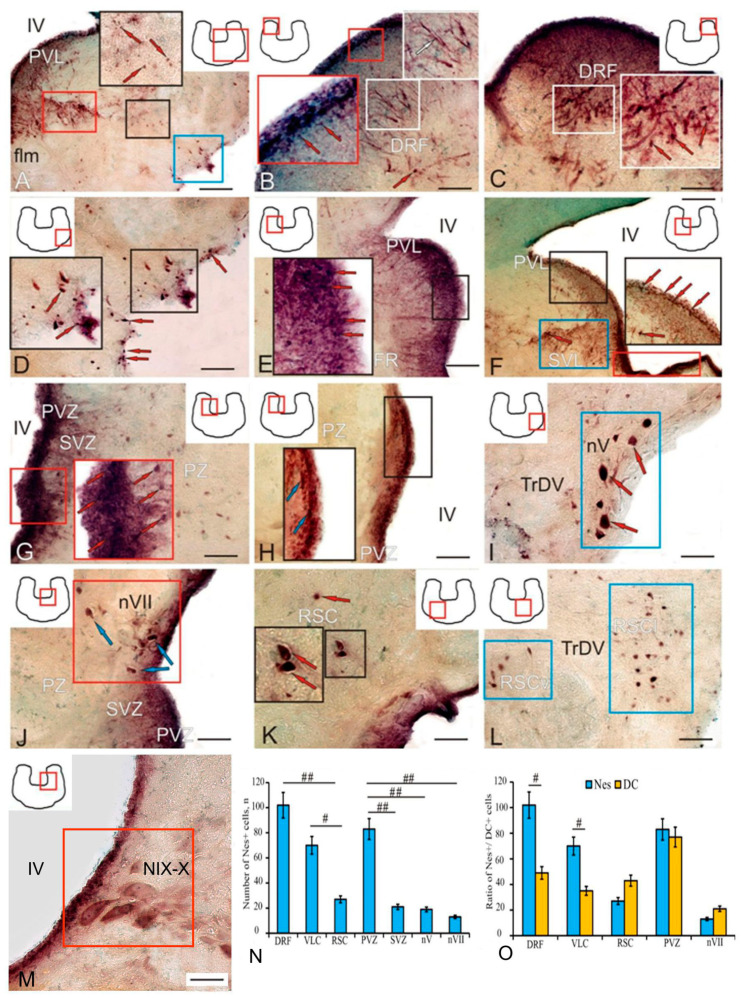
Nestin (Nes) in the brainstem of a rainbow trout, *Oncorhynchus mykiss*. (**A**)—General view of Nes immunolabeling in the lateral reticular formation (LRF, inset, red arrows); Nes+ neuroepithelial (NE) cells in the ventro-lateral zone (VLZ, blue box); Nes+ cells in the medial reticular formation (MRF, red box). (**B**)—Nes+ cells (red arrows) in the marginal zone (red inset); Nes+ RG (white arrow) in the dorsal reticular formation (DRF, white inset). (**C**)—An enlarged fragment of intensely labeled cells (red arrows) and fibers (white inset) in the DRF. (**D**)—Different types of Nes+ cells (red arrows) in the submarginal zone (SMZ, inset). (**E**)—Nes+ cells (red arrows) in the periventricular zone (PVZ) of the lateral wall (inset) of the IV ventricle. (**F**)—Diffuse pattern of distribution of Nes+ NE cells (red arrows) in the PVZ (inset) in the lateral wall of the IV ventricle; immunopositive cells in the subventricular layer (SVL, in blue box) and at the bottom of the IV ventricle (in the red box). (**G**)—An enlarged fragment of immunopositive NE-type cells (red arrows) in the PVZ and SVZ (inset); parenchymal zone (PZ). (**H**)—Intensively labeled cells (blue arrows) located diffusely in the PVZ of the lateral wall of the IV ventricle (inset). (**I**)—Nes+ cells (red arrows) in the lateral nucleus of the V nerve (in blue rectangle). (**J**)—Nes+ cells (blue arrows) in the nucleus of the VII nerve (red square). (**K**)—Nes+ RSC population (inset). (**L**)—Ventral (RSCv) and lateral (RSCl) immunopositive populations (in blue boxes) separated by the descending trigeminal tract. (**M**)—Nes+ neurons (red arrow) in the parasympathetic column and interneurons in the sensory part (white arrow) in the nucleus of IX-X nerves (in the red rectangle). Scale bars: (**A**,**F**,**I**–**L**) 200 µm; (**B**–**E**,**G**,**H**) 100 µm; (**M**)—50 µm. (**N**)—Comparative distribution of Nes+ cells in different areas of the brainstem; significant intergroup differences in the number of cells in the DRF and RSC population ## (*p* < 0.01); the PVZ and SVZ, NV, and NVII ## (*p* < 0.01); and in the VLC and RSC populations # (*p* < 0.05) (*n* = 5 in each group), one-way ANOVA. (**O**)—Comparative distribution of Nes+ and DC+ cells in different areas of the brainstem; significant intergroup differences in the number of Nes+ and DC+ cells in the DRF and VLC population # (*p* < 0.05) (*n* = 5 in each group), one-way ANOVA.

**Figure 11 ijms-25-05595-f011:**
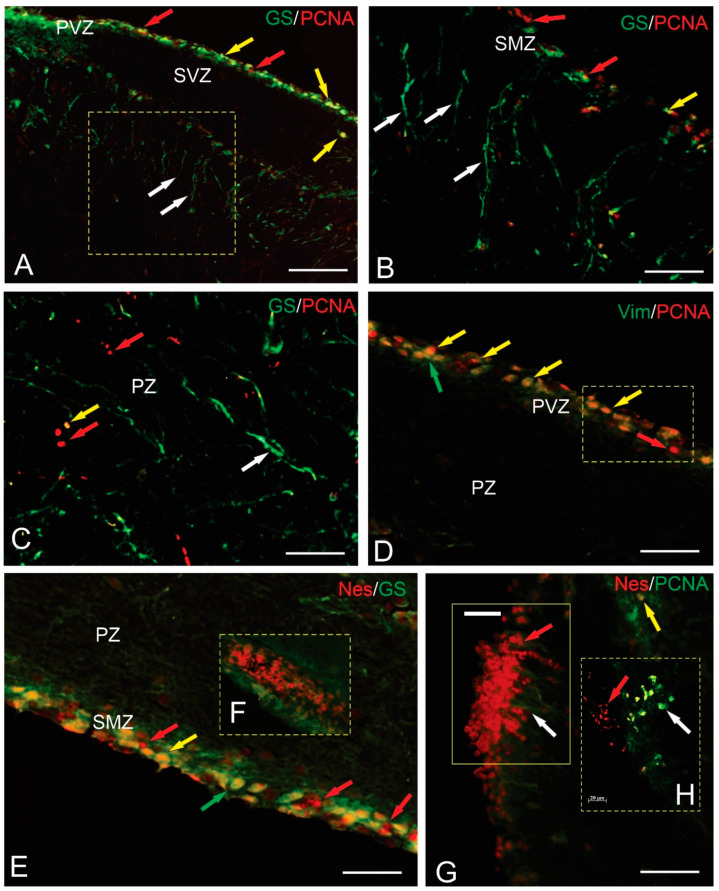
Double immunofluorescence labeling in the brainstem (**A**–**F**) and cerebellum of the trout *Oncorhynchus mykiss*. (**A**)—Patterns of PCNA localization (red arrows) and GS (white arrows) in the PVZ: yellow arrows show cells with PCNA/GS colocalization and yellow dotted lines are limited to GS+ RG fibers; (**B**)—an enlarged fragment of the yellow rectangle in (**A**); (**C**)—a pattern of PCNA and GS immunofluorescence in the parenchymal region (PZ) of the brainstem, notification as in (**A**); (**D**)—patterns of Vim/PCNA colocalization (yellow arrows) in the PVZ (limited by a yellow dotted line) of the brainstem, Vim+ cells (green arrows), and PCNA+ cells (red arrows); (**E**)—patterns of Nes/PCNA colocalization (yellow arrow) in the SMZ of the brainstem, Nes+ cells (red arrows), and PCNA+ cells (green arrows); (**F**)—inset, bounded by yellow dotted line, containing a cluster of Nes+ cells in the SMZ of the brainstem; (**G**)—patterns of Nes/GS immunolocalization in dorso-lateral part of the corpus cerebellum (CC): in a yellow rectangle, a dense cluster of Nes+ cells (red arrows), and in the surface layers of ML, GS+ RG fibers (green arrow); (**H**)—an inset, bounded by a yellow dotted line, containing diffuse accumulation of Nes+/GS+ cells in parenchyma of ML. Scale bar: (**A**,**G**)—200 mkm; (**B**–**D**,**F**)—100 mkm; (**E**)—50 mkm.

**Figure 12 ijms-25-05595-f012:**
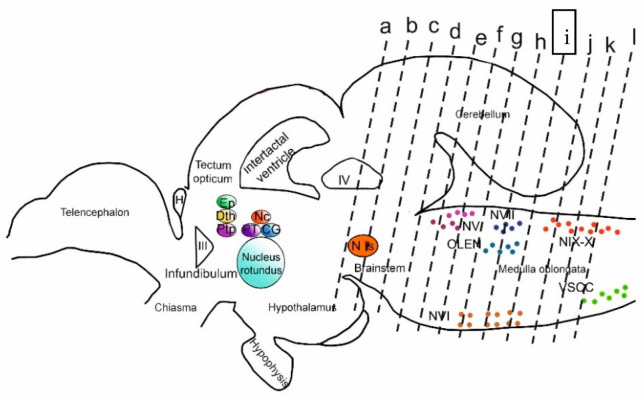
Diagram of organization of the trout brain (sagittal projection) showing the levels of the frontal sections through the cerebellum and brainstem (a–l), for which data on the distribution of glutamine synthetase, vimentin, nestin, and doublecortin are presented. The letter designations are as follows: Ep—epithalamus; Dth—dorsal thalamus; Ptp—posterior tuberal nucleus; PT—posterior tubercle; Nc—cortical nucleus; CG—*corpus geniculatum*; NIs—isthmus nucleus; NV—trigeminal nucleus; NVI—abducens nucleus; NVII—nucleus of the facial nerve; NIX–X—nuclei of the glossopharyngeal and vagus nerves; VSCCs—ventral spinal column of cells.

**Figure 13 ijms-25-05595-f013:**
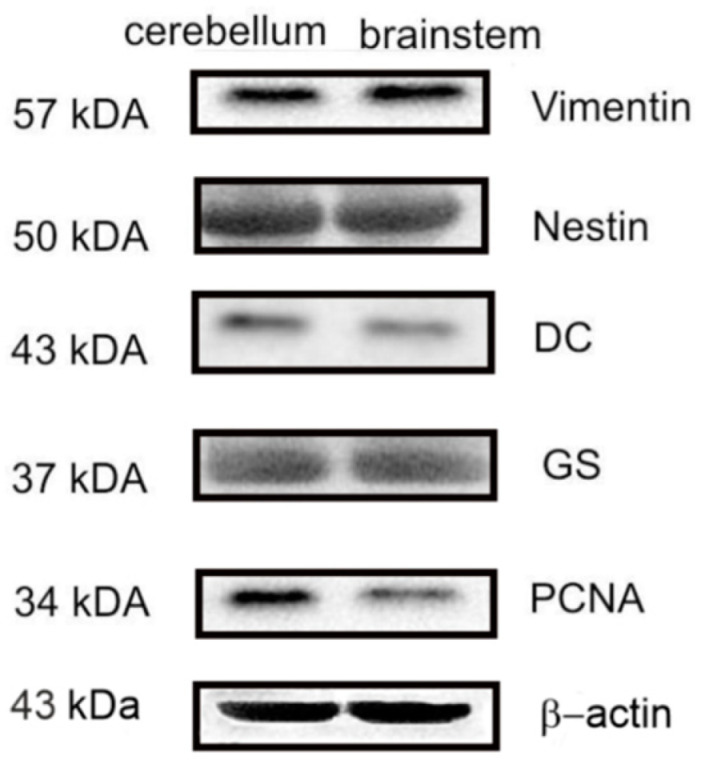
SDS–polyacrylamide gel immunoblots of the adult *Oncorhynchus mykiss* brain protein extracts stained with anti-PCNA, anti-GS, anti-doublecortin, anti-nestin, and anti-vimentin. A single PCNA band corresponding to a molecular weight of 34 kDa in the cerebellum and brainstem was found. The GS lane showed a single band of around 37 kDa. The doublecortin showed a single band around 43 kDa. A single vimentin band corresponding to a molecular weight of around 50 kDa in the cerebellum and brainstem was found. The nestin band showed one band corresponding to 57 kDa. The β-actin band showed one band corresponding to 43 kDa.

**Table 1 ijms-25-05595-t001:** Morphometric and densitometric parameters of proliferating cell nuclear antigen-expressing cells (M ± SD) in the cerebellum and brainstem of rainbow trout, *Oncorhynchus mykiss*.

Brain Area	Type of Cell, Brain Localization	Cell Size	Intensity of Immunolabeling
**Cerebellum**			
Dorso-lateral part rostral area	Undifferentiated (PVZ) Elongated (PZ, ML)	3.9 ± 0.4/2.8 ± 0.6 5.7 ± 0.6/4.3 ± 0.7	+++ +++
Dorsal matrix zone (DMZ)	Undifferentiated (PZ, ML) Elongated (PZ, ML)	4.3 ± 0.4/3.0 ± 0.8 5.4 ± 0.3/3.2 ± 0.6	+++/++ +++
Lateral part	Undifferentiated (PVZ) Elongated (PZ, ML)	4.0 ± 0.4/2.9 ± 0.5 5.6 ± 0.5/4.2 ± 0.5	+++ +++
Basal part	Undifferentiated (PZ, ML) Elongated (PZ, ML)	4.1 ± 0.2/2.7 ± 0.4 5.6 ± 0.5/4.2 ± 0.5	+++ +++
Periventricular zone (PVZ)	Undifferentiated (PVZ) Elongated (ML, GL) Oval (GrL)	4.4 ± 0.5/3.7 ± 0.3 5.8 ± 0.4/3.8 ± 0.8 7.7 ± 1.8/4.2 ± 0.5	+++ +++/++ +++/++
Granular eminences (GrEms)	Undifferentiated (SMZ) Elongated (GrEm) Oval (GrEm)	5.2 ± 0.6/3.6 ± 0.8 6.7 ± 0.4/4.8 ± 0.7 10.1 ± 0.7/6.6 ± 0.5	+++ +++/++ +++/++
**Brainstem**			
Periventricular zone (PVZ)	Undifferentiated Elongated 1 Radial glia 1 Radial glia 2	5.8 ± 0.6/5.4 ± 0.7 7.6 ± 0.6/5.2 ± 0.7 9.4 ± 0.7/5.4 ± 0.9 11.6 ± 0.9/6.6 ± 0.9	+++ +++/++ +++/++ +++/++
Interfascicular zone (IFZ)	Undifferentiated Elongated 1	6.1 ± 0.8/3.5 ± 0.7 7.7 ± 0.8/3.9 ± 1.0	+++/++ +++/++
Submarginal zone (SMZ)	Undifferentiated Elongated 1 (PZ) Radial glia 1	5.3 ± 0.7/3.3 ± 0.8 6.8 ± 0.3/4.1 ± 0.7 8.8 ± 0.2/5.2 ± 0.9	+++ +++ +++/++

Optical density (OD) in immunopositive cells was categorized into high (180–130 UOD, designated as +++) and moderate (130–80 UOD, ++); the initial OD value was measured on the control mounts. Values before slash are for the greater diameter of the cell body; after slash, for the lesser diameter.

**Table 2 ijms-25-05595-t002:** Morphometric and densitometric parameters of glutamine synthetase-expressing cells (M ± SD) in the cerebellum and brainstem of the rainbow trout *Oncorhynchus mykiss*.

Brain Area	Type of Cell	Cell Size, µm	Intensity of Labeling
**Cerebellum**			
Dorso-lateral part	Undifferentiated Elongated 1 Elongated 2 Elongated 3	5.9 ± 0.6/3.7 ± 0.6 7.9 ± 0.3/5.3 ± 0.5 9.7 ± 0.7/5.4 ± 1.2 13.2 ± 1.3/7.6 ± 0.3	+++ +++ +++ +++/++
Dorsal matrix zone (DMZ)	Undifferentiated Elongated	4.5 ± 0.5/3.2 ± 0.6 5.5 ± 0.4/3.2 ± 0.5	+++ +++
Basal part	Undifferentiated Elongated 1 Elongated 2 Elongated 3	6.5 ± 0.2/5.2 ± 0.4 8.4 ± 0.3/6.2 ± 0.6 9.6 ± 0.4/6.9 ± 1.4 12.6 ± 0.5/7.4 ± 2.4	+++ +++ +++/++ +++/++
Baso-lateral part	Undifferentiated Elongated 1 Elongated 2 Elongated 3	6.2 ± 0.8/5.7 ± 1.0 7.6 ± 0.1/4.9 ± 0.3 9.7 ± 0.2/6.3 ± 1.5 11.8 ± 0.6/6.9 ± 1.9	+++ +++ +++/++ +++/++
Periventricular zone (PVZ)	Undifferentiated Elongated 1 Elongated 2 Elongated 3	4.5 ± 0.5/3.7 ± 0.4 7.0 ± 0.5/4.4 ± 0.5 8.4 ± 0.5/4.6 ± 0.6 11.8 ± 0.6/6.4 ± 1.0	+++ +++ +++/++ +++/++
**Brainstem**			
Periventricular zone (PVZ)	Undifferentiated Elongated 1 Radial glia 1 Radial glia 2	6.2 ± 0.4/5.2 ± 1.1 7.9 ± 0.5/5.3 ± 0.6 9.6 ± 0.6/5.2 ± 1.3 13.7 ± 0.7/6.3 ± 1.1	+++/++ +++ +++/++ +++/++
Interfascicular zone (IFZ)	Undifferentiated Elongated 1	6.1 ± 0.8/3.5 ± 0.7 7.7 ± 0.8/3.9 ± 1.0	+++/++ +++/++
Submarginal zone (SMZ)	Undifferentiated Elongated 1 Radial glia 1	5.3 ± 0.7/3.3 ± 0.8 6.8 ± 0.3/4.1 ± 0.7 8.8 ± 0.2/5.2 ± 0.9	+++ +++ +++/++

Optical density (OD) in immunopositive cells was categorized into high (180–130 UOD, designated as +++) and moderate (130–80 UOD, ++); the initial OD value was measured on the control mounts. Values before slash are for the greater diameter of the cell body; after slash, for the lesser diameter.

**Table 3 ijms-25-05595-t003:** Morphometric and densitometric parameters of doublecortin-expressing cells (M ± SD) in the cerebellum and brainstem of the rainbow trout *Oncorhynchus mykiss*.

Brain Area	Type of Cell	Cell Size, µm	Intensity of Labeling
**Cerebellum**			
Basal zone (BZ)	Elongated (ML, GL) Oval (GL) Eurydendroid (GL) Pear-shaped (GL)	11.3 ± 1.6/8.7 ± 1.9 16.2 ± 1.0/10.8 ± 1.7 20.1 ± 1.2/12.4 ± 2.3 24.6 ± 1.6/13.4 ± 1.7	+++ +++ +++/++ +++/++
Dorsal zone (DZ)	Undifferentiated (ML, GL) Elongated 1 (ML, GL) Elongated 2 (ML) Oval (ML) Pear-shaped (GL)	4.9 ± 0.6/4.0 ± 0.4 7.1 ± 0.7/5.2 ± 0.5 8.9 ± 0.9/6.3 ± 1.3 13.9 ± 0.9/8.3 ± 1.9 22.2 ± 3.8/11.8 ± 5.7	+++ +++ +++/++ +++/++ +++/++
Dorsal matrix zone (DMZ)	Undifferentiated (ML, GL) Elongated (ML, GL)	4.2 ± 0.4/3.1 ± 0.55 8 ± 0.8/3.8 ± 0.8	+++ +++
Lateral zone (LZ)	Undifferentiated (ML) Elongated 1 (ML) Oval (ML, GL) Pear-shaped 1 (GL) Pear-shaped 2 (GL)	4.2 ± 0.5/3.0 ± 0.8 8.9 ± 1.2/6.5 ± 2.0 15.4 ± 2.2/11.7 ± 1.2 23.6 ± 1.4/18.7 ± 3.4 26.9 ± 1.2/15.5 ± 2.4	+++ +++/++ +++/++ +++/++ +++/++
Granular layer (Grl)	Oval (GrL) Polygonal 1 (GrL) Polygonal 2 (GrL)	13.2 ± 0.8/8.8 ± 0.8 17.8 ± 2.5/11.0 ± 2.6 23.9 ± 2.3/12.7 ± 2.6	++ +++/++ +++/++
Granular eminence (GrEm)	Elongated (GrEm) Oval 1 (GrEm) Oval 2 (GrEm) Polygonal 1 (GrEm) Polygonal 2 (GrEm)	5.5 ± 0.7/4.2 ± 0.5 10.3 ± 0.2/6.3 ± 0.3 13.3 ± 0.7/8.6 ± 1.7 15.0 ± 0.4/9.0 ± 1.0 18.7 ± 1.0/11.7 ± 1.4	+++/++ ++ +++/++ +++/++ +++/++
**Brainstem**			
Nucleus nervus vagus (NX)	Oval (SVZ) Polygonal 1 (SVZ) Polygonal 2 (SVZ) Polygonal 3 (PZ)	10.7 ± 0.4/8.9 ± 0.5 16.0 ± 0.7/9.2 ± 0.8 22.4 ± 2.7/9.6 ± 1.8 28.9 ± 3.4/11.9 ± 2.7	+++/++ +++/++ +++/++ ++
Nucleus nervus glossopharyngeus (NIX)	Polygonal 1 (SVZ) Polygonal 2 (SVZ) Polygonal 3 (PZ)	19.7 ± 0.9/11.8 ± 0.7 28.5 ± 1.7/16.6 ± 1.4 31.0 ± 2.6/11.6 ± 1.8	++ ++ ++
Interfascicular zone (IFZ)	Oval (IFZ) Polygonal 1 (IFZ) Polygonal 2 (IFZ) Polygonal 3 (IFZ) Polygonal 4 (IFZ)	8.4 ± 0.4/7.2 ± 0.6 14.9 ± 1.8/7.4 ± 3.3 19.5 ± 0.6/12.7 ± 0.4 25.7 ± 1.3/12.8 ± 1.8 31.5 ± 0.7/18.6 ± 4.5	++ ++ +++/++ +++/++ +++/++
Nucleus nervus facialis (NVII)	Polygonal 1 (PZ) Polygonal 2 (PZ) Polygonal 3 (SVZ, PZ) Polygonal 4 (SVZ) Polygonal 5 (PZ) Polygonal 6 (PZ) Polygonal 7 (PZ)	19.7 ± 0.8/11.8 ± 0.7 29.4 ± 1.8/13.4 ± 2.2 40.7 ± 2.3/17.2 ± 3.6 48.2 ± 2.3/27.8 ± 8.0 69.5 ± 4.6/23.7 ± 7.3 80.7 ± 3.7/23.6 ± 2.6 95.4 ± 1.2/20.4 ± 4.5	+++/++ +++/++ +++ ++/+ ++/+ +++/++ ++
Dorsal reticular formation (DRF)	Undifferentiated (PZ) Elongated 1 (PZ) Elongated 2 (PZ) Oval 1 (PZ) Oval 2 (PZ) Polygonal (PZ)	4.2 ± 0.3/3.2 ± 1.4 6.9 ± 0.6/4.5 ± 0.7 8.6 ± 0.5/5.3 ± 0.3 11.0 ± 0.9/8.0 ± 1.0 13.7 ± 0.4/10.5 ± 1.4 21.4 ± 1.2/11.3 ± 2.3	+++ +++ ++/+ ++ +++/++ +++/++
Periventricular layer (PVl) reticulospinal cells (RSCs)	Undifferentiated (PVZ) Elongated (PVZ) Elongated (PVZ, SVZ) Elongated 2 (PZ) Oval (PZ) Polygonal 1 (SVZ, PZ) Polygonal 2 (SVZ) Polygonal 3 (PZ) Polygonal 4 (PZ) Polygonal 5 (PZ) Polygonal 6 (PZ) Polygonal 7 (PZ)	4.4 ± 0.7/3.2 ± 0.5 6.8 ± 0.6/4.5 ± 0.8 8.9 ± 0.6/5.2 ± 0.4 8.4 ± 0.4/7.3 ± 0.6 12.0 ± 1.7/7.5 ± 2.3 15.0 ± 0.3/11.2 ± 0.6 18.3 ± 1.3/11.5 ± 1.9 23.0 ± 1.4/12.2 ± 1.7 27.4 ± 1.3/13.4 ± 4.9 31.2 ± 0.8/17.4 ± 3.6 35.0 ± 1.7/17.4 ± 8.7 41.3 ± 2.5/16.3 ± 2.3	+++ +++ ++/+ +++ +++ +++/++ +++/++ +++/++ ++/+ +++/++ +++/++ +++/++
Ventro-lateral cells (VLCs)	Undifferentiated (PVZ) Elongated 1 (PVZ) Elongated 2 (PVZ, SVZ) Oval (PZ) Polygonal 1 (SVZ, PZ) Polygonal 2 (SVZ)	4.3 ± 0.5/2.7 ± 0.3 7.8 ± 0.9/5.1 ± 0.9 9.9 ± 0.5/6.5 ± 1.5 12.6 ± 0.4/7.5 ± 2.3 15.3 ± 1.0/7.2 ± 1.3 18.9 ± 1.8/11.5 ± 1.6	+++ +++ ++ +++ ++/+ +++/++

Optical density (OD) in immunopositive cells was categorized into high (180–130 UOD, designated as +++), moderate (130–80 UOD, ++), and weak (80–40 UOD, +); the initial OD value was measured on the control mounts. Values before slash are for the greater diameter of the cell body; after slash, for the lesser diameter.

**Table 4 ijms-25-05595-t004:** Morphometric and densitometric parameters of vimentin-expressing cells (M ± SD) in the cerebellum and brainstem of the trout *Oncorhynchus mykiss.*

Brain Area	Type of Cell	Cell Size, µm	Intensity of Labeling
**Cerebellum**			
Dorso-lateral part of rostral area	Undifferentiated (PVZ) Elongated (PZ, ML) Granules (PZ, ML)	3.9 ± 0.4/2.8 ± 0.6 10.9 ± 0.4/7.5 ± 0.4 2.4 ± 0.4/2.4 ± 0.3	+++ +++ +++
Dorsal part of caudal area	Elongated 1 (ML) Elongated 2 (ML) Pear-shaped 1 (GL) Pear-shaped 2 (GL)	7.4 ± 0.2/4.9 ± 0.4 9.5 ± 0.3/6.5 ± 0.6 11.6 ± 0.4/9.3 ± 1.4 13.8 ± 0.5/11.1 ± 2.4	+++ +++ +++/++ +++/++
Dorsal matrix zone DMZ	Undifferentiated (PZ, ML) Elongated (PZ, ML)	4.3 ± 0.4/3.0 ± 0.8 5.4 ± 0.3/3.2 ± 0.6	+++ +++
Lateral part of CC	Undifferentiated (PZ, ML) Granules (PZ, ML)	4.1 ± 0.2/2.7 ± 0.4 2.3 ± 0.4/1.8 ± 0.1	+++ +++
Granular layer (Grl)	Elongated 1 (GrL) Elongated 2 (GrL) Oval (GrL)	6.2 ± 0.8/5.2 ± 1.3 8.0 ± 0.7/6.0 ± 0.5 9.6 ± 0.5/6.9 ± 0.6	+++ +++ +++
Periventricular zone (PVZ)	Granules (PVZ)	2.4 ± 0.3/2.2 ± 0.3	+++
Granular eminence (GrEm)	Undifferentiated (SMZ) Elongated (GrEm) Oval (GrEm) Polygonal 1 (GrEm) Polygonal 2 (GrEm)	5.2 ± 0.6/3.6 ± 0.8 6.7 ± 0.4/4.8 ± 0.7 10.3 ± 0.8/6.8 ± 0.6 12.2 ± 1.1/8.4 ± 3.5 16.1 ± 0.4/11.4 ± 1.7	+++ +++/++ +++/++ +++/++ +++/++
**Brainstem**			
Periventricular zone (PVZ)	Undifferentiated (PVZ) Elongated 1 (PVZ) Elongated 2 (PVZ) Radial glia 1 (PVZ) Radial glia 2 (PVZ)	5.2 ± 0.4/3.3 ± 0.5 6.5 ± 0.3/3.7 ± 0.7 7.5 ± 0.3/4.0 ± 0.6 8.7 ± 0.5/5.9 ± 1.2 10.6 ± 0.7/4.5 ± 0.3	+++ +++ +++/++ +++/++ +++/++
Principal and lateral nucleus of nervi trigeminalis (NV)	Polygonal 1 (PZ) Polygonal 2 (PZ) Polygonal 3 (SVZ) Polygonal 4 (PZ) Polygonal 5 (PZ)	19.3 ± 1.4/18.9 ± 1.6 27.0 ± 2.0/16.7 ± 4.6 40.6 ± 2.0/24.4 ± 7.5 46.3 ± 0.9/24.2 ± 10.0 54.3 ± 1.5/30.4 ± 4.0	+++/++ +++/++ +++/++ +++/++ +++/++
Nucleus nervus facialis (NVII)	Polygonal 1 (PZ) Polygonal 2 (PZ) Polygonal 3 (SVZ) Polygonal 4 (PZ) Polygonal 5 (PZ) Polygonal 4 (PZ) Polygonal 5 (PZ)	24.4 ± 2.1/14.8 ± 3.3 28.7 ± 1.5/19.2 ± 4.4 35.3 ± 2.8/20.9 ± 6.7 42.2 ± 0.7/15.6 ± 3.0 47.8 ± 1.5/23.4 ± 1.8 52.9 ± 1.9/22.8 ± 5.8 74.5 ± 3.5/19.5 ± 4.0	+++/++ +++/++ +++/++ +++/++ +++/++ +++/++ +++/++
Nucleus nervus vagus (NX)	Oval (SVZ) Polygonal 1 (SVZ) Polygonal 2 (SVZ) Polygonal 3 (PZ) Polygonal 4 (PZ)	11.4 ± 0.3/7.8 ± 0.5 16.0 ± 0.9/9.5 ± 0.7 21.5 ± 3.4/10.2 ± 2.5 30.2 ± 4.2/12.7 ± 2.7 41.9 ± 5.2/16.5 ± 3.2	++ ++ +++/++ +++ +++
Lateral reticular formation (LRF)	Polygonal 1 (PZ) Polygonal 2 (PZ) Polygonal 3 (PZ) Polygonal 4 (PZ) Polygonal 5 (PZ)	14.7 ± 0.4/7.7 ± 0.6 20.5 ± 0.5/14.3 ± 3.4 27.4 ± 1.1/17.0 ± 3.1 31.1 ± 2.9/16.8 ± 2.1 56.1 ± 6.6/32.0 ± 8.1	+++/++ +++/++ +++/++ +++/++ +++/++
Medial reticular formation (MRF)	Polygonal 3 (SVZ) Polygonal 4 (PZ) Bipolar (PZ)	28.3 ± 2.1/14.6 ± 2.4 35.3 ± 3.1/17.4 ± 2.8 46.4 ± 5.2/19.3 ± 7.2	+++/++ +++/++ +++/++
Reticulospinal cells (RSCs)	Polygonal 2 (IFZ) Polygonal 3 (IFZ) Polygonal 4 (IFZ) Polygonal 5 (IFZ)	31.7 ± 3.9/24.0 ± 1.5 38.6 ± 1.3/23.3 ± 8.6 45.6 ± 2.2/30.8 ± 0.9 56.2 ± 3.6/27.6 ± 5.1	+++/++ +++/++ +++/++ +++/++
Submarginal zone (SMZ)	Undifferentiated (PVZ) Undifferentiated (PVZ) Elongated (PVZ)	3.4 ± 0.3/2.3 ± 0.4 5.2 ± 0.6/3.4 ± 0.7 6.8 ± 0.6/3.6 ± 0.4	+++ +++ +++/++

Optical density (OD) in immunopositive cells was categorized into high (180–130 UOD, designated as +++) and moderate (130–80 UOD, ++); the initial OD value was measured on the control mounts. Values before slash are for the greater diameter of the cell body; after slash, for the lesser diameter.

**Table 5 ijms-25-05595-t005:** Morphometric and densitometric parameters of nestin-expressing cells (M ± SD) in the cerebellum and brainstem of the rainbow trout *Oncorhynchus mykiss*.

Brain Area	Type of Cell, Brain Localization	Cell Size, µm	Intensity of Labeling
**Cerebellum**			
Dorsal zone of molecular layer (DZ)	Undifferentiated (ML, GL) Elongated 1 (ML, GL) Elongated 2 (ML) Oval (ML)	4.2 ± 0.4/3.1 ± 0.5 5.7 ± 0.6/4.3 ± 0.7 8.0 ± 0.5/5.2 ± 1.6 11.4 ± 0.8/6.4 ± 1.5	+++ +++ +++/++ +++/++
Dorsal zone of Purkinje cells (PCs)	Pear-shaped (GL) Pear-shaped (GL) Pear-shaped (GL) Pear-shaped (GL) Pear-shaped (GL)	18.2 ± 1.2/11.6 ± 1.6 23.0 ± 1.8/14.6 ± 1.4 27.6 ± 1.9/16.4 ± 3.4 23.6 ± 1.4/18.7 ± 3.4 31.5 ± 1.8/11.6 ± 2.4	+++/++ +++/++ +++/++ +++/++ +++/++
Dorsal matrix zone (DMZ)	Undifferentiated (ML, GL) Elongated (ML, GL)	4.1 ± 0.6/2.5 ± 0.5 5.7 ± 0.5/3.2 ± 0.4	+++ +++
Lateral zone (LZ) of Purkinje cells (PCs)	Pear-shaped (GL) Pear-shaped (GL) Pear-shaped (GL) Pear-shaped (GL)	16.6 ± 2.2/13.1 ± 1.2 20.0 ± 1.5/15.7 ± 3.2 24.1 ± 1.2/12.8 ± 2.3 28.9 ± 0.8/13.3 ± 1.7	+++/++ +++/++ +++/++ +++/++
Basal zone (BZ) of Purkinje cells (PCs)	Pear-shaped (GL) Pear-shaped (GL) Pear-shaped (GL) Pear-shaped (GL) Pear-shaped (GL)	14.3 ± 1.3/11.5 ± 2.7 17.9 ± 0.4/14.3 ± 2.5 18.9 ± 0.9/12.9 ± 1.3 22.2 ± 0.7/10.6 ± 2.5 26.0 ± 0.9/11.7 ± 1.0	+++/++ +++/++ +++/++ +++/++ +++/++
Periventricular zone (PVZ)	Undifferentiated (PVZ) Elongated (ML, GL) Oval (GrL)	4.4 ± 0.5/3.7 ± 0.3 5.8 ± 0.4/3.8 ± 0.8 7.7 ± 1.8/4.2 ± 0.5	++ +++/++ +++/++
Granular eminence (GrEm)	Undifferentiated (PVZ) Elongated (GrEm) Oval (GrEm)	4.7 ± 0.8/3.3 ± 0.5 6.8 ± 0.6/4.6 ± 1.0 12.3 ± 0.7/7.5 ± 1.7	+++/++ ++ +++/++
**Brainstem**			
Dorsal reticular formation (DRF)	Undifferentiated (PZ) Elongated 1 (PZ) Elongated 2 (PZ) Oval 1 (PZ) Oval 2 (PZ) Polygonal (PZ)	4.5 ± 0.4/3.2 ± 0.4 6.7 ± 0.4/4.3 ± 0.7 8.4 ± 0.5/5.3 ± 0.3 11.2 ± 0.7/7.7 ± 1.2 14.0 ± 0.6/11.3 ± 1.6 20.2 ± 1.6/11.4 ± 2.3	+++ +++ +++/++ +++/++ +++/++ +++/++
Nucleus nervus trigeminalis (NV)	Polygonal 1 (PZ) Polygonal 2 (PZ) Polygonal 3 (SVZ, PZ) Polygonal 4 (SVZ) Polygonal 5 (PZ)	31.3 ± 1.8/21.9 ± 2.7 44.9 ± 2.6/32.0 ± 3.1 57.2 ± 2.8/32.4 ± 6.6 60.9 ± 5.6/28.2 ± 5.6 74.8 ± 5.8/42.4 ± 4.6	+++/++ +++/++ +++ +++/++ +++/++
Nucleus nervus facialis (NVII)	Polygonal 1 (PZ) Polygonal 2 (PZ) Polygonal 3 (SVZ, PZ)	29.4 ± 2.6/19.6 ± 3.4 41.5 ± 2.8/29.7 ± 4.7 52.7 ± 3.4/35.1 ± 5.8	+++/++ +++/++ +++
Nucleus nervus vagus (NX)	Oval (SVZ) Polygonal 1 (SVZ) Polygonal 2 (SVZ) Polygonal 3 (PZ) Polygonal 4 (PZ)	11.6 ± 0.7/8.3 ± 0.6 15.2 ± 1.8/9.8 ± 0.8 24.3 ± 4.6/12.3 ± 3.2 36.3 ± 5.1/13.4 ± 3.8 42.7 ± 6.3/17.2 ± 4.7	++ ++ +++/++ +++/++ +++/++
Periventricular layer (PVL)	Undifferentiated (PVZ) Elongated 1 (PVZ) Elongated 2 (PVZ)	4.9 ± 0.5/3.3 ± 0.5 6.7 ± 0.5/4.3 ± 0.4 8.6 ± 0.5/5.0 ± 0.7	+++ +++ +++/++
Reticulospinal cells (RSCs)	Elongated (PVZ, SVZ) Elongated 2 (PZ) Oval (PZ) Polygonal 1 (SVZ, PZ) Polygonal 2 (SVZ) Polygonal 3 (PZ) Polygonal 4 (PZ) Polygonal 5 (PZ) Polygonal 6 (PZ) Polygonal 7 (PZ)	8.9 ± 0.6/5.2 ± 0.4 8.4 ± 0.4/7.3 ± 0.6 12.0 ± 1.7/7.5 ± 2.3 15.0 ± 0.3/11.2 ± 0.6 18.3 ± 1.3/11.5 ± 1.9 23.0 ± 1.4/12.2 ± 1.7 27.4 ± 1.3/13.4 ± 4.9 31.2 ± 0.8/17.4 ± 3.6 35.0 ± 1.7/17.4 ± 8.7 41.3 ± 2.5/16.3 ± 2.3	+++/++ +++ +++ +++/++ +++/++ +++/++ +++/++ +++/++ +++/++ +++/++
Ventro-lateral external cells (VLCs)	Undifferentiated (PVZ) Elongated (PVZ) Elongated (PVZ, SVZ) Oval (PZ) Polygonal 1 (SVZ, PZ) Polygonal 2 (SVZ)	4.3 ± 0.5/2.7 ± 0.3 7.8 ± 0.9/5.1 ± 0.9 9.9 ± 0.5/6.5 ± 1.5 12.6 ± 0.4/7.5 ± 2.3 15.3 ± 1.0/7.2 ± 1.3 18.9 ± 1.8/11.5 ± 1.6	+++ +++ +++/++ +++ +++/++ +++/++

Optical density (OD) in immunopositive cells was categorized into high (180–130 UOD, designated as +++) and moderate (130–80 UOD, ++); the initial OD value was measured on the control mounts. Values before slash are for the greater diameter of the cell body; after slash, for the lesser diameter.

## Data Availability

Data are contained within the article and Appendix A.

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
