# Peer review of "Constitutive Neurogenesis and Neuronal Plasticity in the Adult Cerebellum and Brainstem of Rainbow Trout, Oncorhynchus mykiss"

_ijms, 2024, doi:10.3390/ijms25115595_

Round 1
Reviewer 1 Report
Comments and Suggestions for Authors
The article titled "Constitutive Neurogenesis and Neuronal Plasticity in the Adult Cerebellum and Brainstem of Rainbow Trout, Oncorhynchus" seeks to elucidate the intricate molecular mechanisms governing proliferation, neurogenesis, and neuronal plasticity within two pivotal regions of the central nervous system in Rainbow Trout.
Through a meticulously conducted study, the authors provided a comprehensive analysis of the cerebellum and brainstem in adult rainbow trout, Oncorhynchus mykiss. Their investigation shed light on the spatial distribution of proliferative activity and the expression patterns of glutamine synthetase, vimentin, nestin and doublecortin immunolabeling of neurons originating in the postembryonic period.
Recognizing the translational potential of their findings to mammalian models, the authors underscored the relevance of understanding these molecular mechanisms across various diseases and therapeutic avenues, spanning from regenerative medicine to cancer therapy.
The conclusions drawn succinctly encapsulated the empirical evidence garnered from their study. Additionally, the references cited were aptly selected, reflecting contemporary literature pertinent to the research domain.
However, the authors are encouraged to consider several enhancements for their study:
- Clarification regarding the presentation of data within tables, elucidating whether reported values represent means and standard deviations.
- Inclusion of original SDS-polyacrylamide gel immunoblots depicting adult Oncorhynchus mykiss brain protein extracts stained with anti-PCNA, anti-GS, anti-Doublecortin, anti-Nestin, and anti-Vimentin, within the materials and methods section.
- Further elaboration on the statistical methodologies employed, including rationale for selecting parametric analyses over non-parametric alternatives. Additionally, verification of data distribution, possibly through an assessment of normality, is recommended and should be integrated into the methodology section.
Author Response
The article titled "Constitutive Neurogenesis and Neuronal Plasticity in the Adult Cerebellum and Brainstem of Rainbow Trout, Oncorhynchus" seeks to elucidate the intricate molecular mechanisms governing proliferation, neurogenesis, and neuronal plasticity within two pivotal regions of the central nervous system in Rainbow Trout.
Through a meticulously conducted study, the authors provided a comprehensive analysis of the cerebellum and brainstem in adult rainbow trout, Oncorhynchus mykiss. Their investigation shed light on the spatial distribution of proliferative activity and the expression patterns of glutamine synthetase, vimentin, nestin and doublecortin immunolabeling of neurons originating in the postembryonic period.
Recognizing the translational potential of their findings to mammalian models, the authors underscored the relevance of understanding these molecular mechanisms across various diseases and therapeutic avenues, spanning from regenerative medicine to cancer therapy.
The conclusions drawn succinctly encapsulated the empirical evidence garnered from their study. Additionally, the references cited were aptly selected, reflecting contemporary literature pertinent to the research domain.
However, the authors are encouraged to consider several enhancements for their study:
- Clarification regarding the presentation of data within tables, elucidating whether reported values represent means and standard deviations.
Thank you for your comment, it has been added for all tables that the data is presented as an average ± standard deviation
- Inclusion of original SDS-polyacrylamide gel immunoblots depicting adult Oncorhynchus mykiss brain protein extracts stained with anti-PCNA, anti-GS, anti-Doublecortin, anti-Nestin, and anti-Vimentin, within the materials and methods section.
A selection of the original gels has been added and is presented in the original immunoblots folder.
- Further elaboration on the statistical methodologies employed, including rationale for selecting parametric analyses over non-parametric alternatives. Additionally, verification of data distribution, possibly through an assessment of normality, is recommended and should be integrated into the methodology section.
Parametric analysis methods were used in the work, in particular the Student-Newman-Keuls test, as well as single- and double-sided analysis of variance, since they have greater power compared with nonparametric criteria, as well as the quantitative nature of variation and normal distribution were present in the studied brain structures. The normality of the distribution was tested using a normal probability graph.

Reviewer 2 Report
Comments and Suggestions for Authors
This article is relatively reasonable and reliable. The article researched cerebellum and brainstem of Oncorhynchus mykiss under conditions of homeostatic growth in adult using immunohistochemical methods and Western Immunobloting.The review’s logic is rigorous, and the data of the experiments are convincing. The article assessed the distribution of proliferative activity and GS, Vim, and Nes, as well as DC immunolabeling of neurons formed in the postembryonic period, in the cerebellum and brainstem of adult rainbowtrout, Oncorhynchus mykiss, under conditions of homeostatic growth. It is a topic of interest to the researchers in the related areas, but the article still has some problems.
1. Add a description of the promising applications of this research in the discussion section.The article lacks advantages for the diagnosis of psychiatric disorders by exosomes.
2. The table of criptions are not clear enough and are difficult to navigate.
3. Every specific professional terminology in this article should be list the whole noun at first although in abstract.
4. A flowchart was proper to add in this article because of the complex mechanism.
Author Response
This article is relatively reasonable and reliable. The article researched cerebellum and brainstem of Oncorhynchus mykiss under conditions of homeostatic growth in adult using immunohistochemical methods and Western Immunobloting.The review’s logic is rigorous, and the data of the experiments are convincing. The article assessed the distribution of proliferative activity and GS, Vim, and Nes, as well as DC immunolabeling of neurons formed in the postembryonic period, in the cerebellum and brainstem of adult rainbowtrout, Oncorhynchus mykiss, under conditions of homeostatic growth. It is a topic of interest to the researchers in the related areas, but the article still has some problems.
- Add a description of the promising applications of this research in the discussion section. The article lacks advantages for the diagnosis of psychiatric disorders by exosomes.
Thanks for the comment; a section on the prospects of applying this study has been added to the Conclusion section. However, the comment on the benefits of exosomes for the diagnosis of mental disorders regarding this work is not clear.
- The table of criptions are not clear enough and are difficult to navigate.
The article presents 5 Tables in which morphometric, densitometric and topographic characteristics of immunopositive cells are given. All descriptions related to the explanation of quantitative data are given in the Results section.
- Every specific professional terminology in this article should be list the whole noun at first although in abstract.
All professional terms are deciphered at the first mention in the text. In addition, a list of neuroanatomic Abbreviations used is provided at the end of the work.
- A flowchart was proper to add in this article because of the complex mechanism.
Thank you for this recommendation, however, given the large amount of data obtained, in particular, several cell types were identified for each IHC marker in different areas of the cerebellum and brainstem, in this paper we would like to limit ourselves to detailed documentation and presentation of these data, as well as a discussion of the identified areas in the caudal parts of the trout brain with already known areas in similar parts of the brain of other fish. Unfortunately, it is currently somewhat premature to discuss the mechanisms by which various types of cells with neuron- and gliospecific markers work in the causal parts of the trout brain. We would prefer to discuss these mechanisms in our future research.

Reviewer 3 Report
Comments and Suggestions for Authors
In the article entitled "Constitutive Neurogenesis and Neuronal Plasticity in Adult Cerebellum and Brainstem of Rainbow Trout, Oncorhynchus mykiss" the authors elegantly show adult neurogenesis in the cerebellum and brainstem of rainbow trout through the distribution of progenitor neural stem cells which express GS, Vim and Nes.
It is a very complete study where neurogenic zones in different brain regions are studied. The manuscript is well described and well developed with detailed immunohistochemistry.
Only one doubt.
Throughout the manuscript on page 7 and 9 line 211, 213, 249 and 251 respectively you mention PSNA + cells, please define those cells.
Author Response
In the article entitled "Constitutive Neurogenesis and Neuronal Plasticity in Adult Cerebellum and Brainstem of Rainbow Trout, Oncorhynchus mykiss" the authors elegantly show adult neurogenesis in the cerebellum and brainstem of rainbow trout through the distribution of progenitor neural stem cells which express GS, Vim and Nes.
It is a very complete study where neurogenic zones in different brain regions are studied. The manuscript is well described and well developed with detailed immunohistochemistry.
Only one doubt.
Throughout the manuscript on page 7 and 9 line 211, 213, 249 and 251 respectively you mention PSNA + cells, please define those cells.
Thanks for this comment, the designations of moderately immuno-labeled cells with white arrows have been added to the 2D figure.
